

# Ensemble Projection of the Sea Level Rise Impact on Storm Surge
# and Inundation in the Coastal Bangladesh
Mansur Ali Jisan[1], Shaowu Bao[1], Leonard J. Pietrafesa[1]
[1]Department of Coastal and Marine Systems Science, Coastal Carolina University,  Conway, South Carolina, United States.
*Correspondence to*: Mansur Ali Jisan (mjisan@g.coastal.edu)
**Abstract.**
The hydrodynamic model Delft3D is used to study the impact of Sea Level Rise (SLR) on storm surge and inundation in the
coastal region of Bangladesh. To study the present day inundation scenario, track of two known tropical cyclones (TC) were
used: Aila (Category 1; 2009) and Sidr (Category 5; 2007). Model results were validated with the available observations. Future
inundation scenarios were generated by using the strength of TC Sidr, TC Aila and an ensemble of historical TC tracks but
incorporating the effect of SLR.
Since future change in storm surge inundation under SLR impact is a probabilistic incident, that's why a probable range of future
change in inundated area was calculated by taking in to consideration the uncertainties associated with TC tracks, intensities and
landfall timing.
The model outputs showed that, the inundated area for TC Sidr, which was calculated as 1860 km$^2$, would become 31% higher
than the present day scenario if a SLR of 0.26 meter occurs during the mid-21$^{st}$ century climate scenario. Similar to that, an
increasing trend was found for the end of the 21$^{st}$ century climate scenario. It was found that with a SLR of 0.54 meter, the
inundated area would become 53% higher than the present day case.
Along with the inundation area, the impact of SLR was examined for the changes in future storm surge level. A significant
increase of 21% was found in storm surge level for the case of TC Sidr in Barisal station if a Sea Level Rise of 0.26 meter occurs
at the middle of the 21$^{st}$ century. Similar to that, an increase of 37% was found in storm surge level with a SLR of 0.54 meter in
this location for the end of the 21$^{st}$ century climate scenario.
Ensemble projections based on uncertainties of future TC events also showed that, for a change of 0.54 meters in SLR, the
inundated area would range between 3500-3750 km$^2$ whereas for present day SLR simulations it was found within the range of
1000-1250 km$^2$
These results revealed that even if the future TCs remain at the same strength as at present, the projected changes in SLR will
generate more severe threats in terms of surge height and extent of inundated area.









## 1. Introduction

In addition to routine inundation from upstream river water and the downstream tides, the coastal part of Bangladesh is frequently flooded by storm surges induced by tropical cyclones (TCs). Typically, TC-induced storm surges in this area initiate in the central or southern part of the Bay of Bengal or near the Andaman Sea. TCs normally occur during April – May, the pre-monsoon period, and again from October – November, the post monsoon period. Harris (1963) mentioned that five basic processes (i.e., pressure, direct wind, earth's rotation, waves and rainfall effects) cause water level rise under storm conditions. Pietrafesa *et al.* (1986) also pointed out that high water at the mouths of coastal estuaries, bays and rivers can block discharges of upstream waters and contribute to upstream flooding; a non-local effect. Among these processes, storm surges form primarily due to the TC wind stresses mechanically driving the surface frictional layer onshore. Assuming an idealized balance between pressure gradient force and surface wind stress with assumed small bottom stress, the surge related to TC wind stress can be expressed as $Dh = \frac{t_w L}{g \rho h}$, where $L$ is the fetch of the wind (the distance over which the wind blows), $\tau_w$ is the wind stress due to the friction between the moving air and water surface, $g$ is the gravity, $\rho$ is the density of water, $h$ is the depth near the coast (Hearn, 2008). Also, as a secondary process, due to the differences in pressure level, the water level rises in the areas of low atmospheric pressure and falls in the areas of high atmospheric pressure, which is how the rising water level offsets the low atmospheric pressure to keep the total pressure constant (Harris *et al.*, 1963).

According to Murty et al. (1986), the surge amplifies as it approaches the coast due to the shallow continental shelf of the Bay of Bengal and hence it causes massive flooding in the low-lying coastal areas. A large percentage of the Bangladesh population resides in the low lying coastal regions of the country. Most of the areas near the coastal zone of Bangladesh have been formed by the process of riverine sedimentation and because of that the low lying areas are relatively flat and as such are susceptible to flooding even under normal astronomical tide conditions. Furthermore, the triangular shape of the Bay of Bengal region makes storm surges more distressing, as a funneling effect occurs. The geomorphological characteristics of the region have made the locale prone to major TC events, events which have occurred multiple times in the past, directly causing the deaths of hundreds of thousands of lives (Haque, 1997). This type of coastal flooding associated with the changes in coastal water level due to storms passing over the sea causes great loss of human lives, property, livelihoods and the economy of the country (Haque, 1997).

Future climate change scenarios may further exacerbate the threats of TC-induced storm surge and inundation. According to the Intergovernmental Panel on Climate Change Fourth Assessment Report (IPCC 4AR), there is a high probability of major changes in TC activity across various ocean basins including the Arabian Sea and the Bay of Bengal. According to Milliman et al. (1989), this Ganges-Brahmaputra-Meghna Delta region has long been characterized as a highly vulnerable zone due to its exposure to the increasing trend of SLR. According to the SLR analysis done by the South Asian Association for Regional Cooperation based on the 22-year records of observed sea level at Charchanga, Cox's Bazar and Hiron Point, sea level is rising at rates of 6.0, 7.8 and 4 mm/year, respectively in those three locations (SMRC, 2003). These rates are much higher than the global rate of SLR (~ 3.2 mm/year) over the last 25 years (Pietrafesa et al., 2016). Based on Warrick et al. (1996), the sea level in the Bay of Bengal is also influenced by local factors including tectonic setting, deltaic processes and sediment load; for example, the coastal region of Bangladesh has been subsiding due to the pressure on the Earth's crust from the sediment with thick layers that has formed over millions of years. Warrick et. al. (1996) also analyzed the recent history of land accretion and suggested that the subsidence is also balanced by land accretion due to sediment supply from the coast. These physical phenomena have been



shaping the coast of Bangladesh over the past 100 years. A global SLR of 26-59 cm has been projected over the next 84 years to
2100 by the IPCC under the scenario A1F1 (Meehl et al. 2007). In this proposed work, we will use the SLR projections from
Caesar et al. (2017; under review), which suggests a projection of SLR of 26 cm for the mid-21st century (2040 -2060) and 54
cm for the end-21st century (2079 -2099).
Previous studies have analyzed the likely impact of climate change, especially SLR, on storm surge and inundation in this region.
Using hydrodynamic models, Ali (1992) showed that with an increase of 1.0 and 1.5 meters of SLR, 10% and 15.5%,
respectively, of the entirety of Bangladesh would get flooded under the strength of future TCs. Karim and Mimura (2008) used a
1-D hydrodynamic model to study the inundation under several scenarios of climate and for the case of future TCs by changing
sea surface temperature, SLR, wind speed and sea level pressure. Based on their results, Karim and Mimura (2008) concluded
that with an increase of 2°C in SST and 0.3 meters of SLR, the flood risk area would be 15.3% more than the present day risk
area and the depth of flooding would increase by as much as 22.7% within 20 km from the coastline. Both Ali (1992) and Karim
& Mimura (2008) considered SST rise and future strength of TCs in simulating the future storm surge and inundation.
However, the impact of climate change on the frequency and intensity of TCs are still debatable (Knutson *et al.*, 2010). The
projection of the TC characteristics in the Bay of Bengal region is unclear as well. To improve these uncertainties, a reasonable
method to examine the impact of future SLR on storm surge and inundation would be to construct an ensemble of tracks and
intensities of possible land-falling TCs along the Bangladesh coast based on the historical TC records. From this statistical
approach, we can quantify the probable impact of TC tracks under future SLR change. To date, such an approach has not been
done and will be method of this study.
We first use Delft3D to simulate the present day storm surge and inundation using the strength of two recent TCs (TC Sidr and
TC Aila), and validate the simulations with observational data. Future storm surge and inundation scenarios were then generated
by incorporating the projected SLR.
The study was carried out in the Ganges-Brahmaputra-Meghna Delta regions (Figure 1). According to Integrated Coastal Zone
Management Plan, 19 districts of Bangladesh located near the Bay of Bengal area were defined as the coastal areas. We've
considered all those in this study. We selected two TC cases in this study, a strong Saffir-Simpson(SS) Category-5 that directly
hit the study area, TC Sidr, and the other a SS Category -1 storm that made landfall in the South-West part of the study domain,
TC Aila.
TC Sidr made landfall near Barguna district (Figure 1) in 2007, causing ~ 3000 human fatalities and leaving millions homeless.
This category-5 cyclone is considered one of the most powerful cyclones in the past 15 years to have made landfall in
Bangladesh, which affected over nine millions people living across the Bangladesh coastal areas. The districts of Patuakhali,
Khulna, Barguna and Jhalokathi were badly affected. During TC Sidr, around 15% of the affected people took refuge in nearby
cyclone shelters. In the village of Angul Kata in Barguna district, around 1500 people took shelter in eight reinforced pillars to
protect themselves from the tidal surge of around 5 meters. If there had been no shelters, the death toll could have reached into
the hundreds in that area.
The other cyclone studied in this paper, TC Aila (Figure 1) occurred in the Bay of Bengal region in 2009. Although a category-1
storm, Aila caused ~ 190 deaths and affected 4.8 million people, the devastation that left a long term impact. The locales mainly
affected were Khulna, Patukhali and Chandpur. The storm surge due to Aila broke a dam in Pataukhali and submerged five
villages, destroying a huge number of homes and leaving thousands of people homeless. Most of the people living in those





affected areas took shelter in the nearest cyclone shelters. According to government sources, approximately 2,500,000 houses
had been destroyed completely and 3,700,000 houses had been damaged.
TC Sidr made landfall near Barguna district (Figure 1) in 2007, causing ~ 3000 human fatalities and leaving millions homeless.
TC Aila occurred in the Bay of Bengal region in the year 2009 (Figure 1). Although a category-1 storm, Aila caused ~ 190
deaths and affected 4.8 million people, devastation that left a long term impact. The locales mainly affected were Khulna,
Patukhali and Chandpur. The storm surge due to Aila broke a dam in Pataukhali and submerged five villages, destroying huge
number of homes and leaving thousands of people homeless.
**2. Methodology**
**2.1 Modeling Methodology**
**2.1.1 Application of Numerical Model**

For the purpose of developing present day and future inundation scenario in the coastal regions of Bangladesh, the Delft3D-
FLOW (Delft Hydraulics, 2006), a multidimensional (2D or 3D) hydrodynamic and transport simulation program that calculates
non-steady flow and transport phenomena resulting from tidal and meteorological forcing was used. Delft3D-FLOW solves the
unsteady shallow water equation in two dimensions (depth-averaged) or in three dimensions. The system of equations consists of
the horizontal equations of motion, the continuity equation, and the transport equations for conservative constituents. The
equations are formulated in orthogonal curvilinear co-ordinates or in spherical co-ordinates. Delft3D – FLOW module's two-
dimensional, depth averaged flow equations can be applied for modeling tidal waves, storm surges, tsunamis, harbor oscillations
(seiches) and transport of pollutants in vertically well-mixed flow regimes. In this paper Delft3D's 2D mode for barotropic
depth-integrated flow has been applied. The equations are listed below.
$$\frac{\partial \zeta}{\partial t} + \frac{1}{\sqrt{G_{\xi\xi}}\sqrt{G_{\eta\eta}}} \frac{\partial\left[(d+\zeta)v\sqrt{G_{\eta\eta}}\right]}{\partial \xi} + \frac{1}{\sqrt{G_{\xi\xi}}\sqrt{G_{\eta\eta}}} \frac{\partial\left[(d+\zeta)v\sqrt{G_{\xi\xi}}\right]}{\partial \xi} = Q \tag{1}$$

$$\frac{\partial u}{\partial t} + \frac{u}{\sqrt{G_{\xi\xi}}}\frac{\partial u}{\partial \xi} + \frac{v}{\sqrt{G_{\eta\eta}}}\frac{\partial u}{\partial \eta} + \frac{uv}{\sqrt{G_{\xi\xi}}\sqrt{G_{\eta\eta}}}\frac{\partial \sqrt{G_{\xi\xi}}}{\partial \eta} - \frac{v^2}{\sqrt{G_{\xi\xi}\sqrt{G_{\eta\eta}}}}\frac{\partial \sqrt{G_{\eta\eta}}}{\partial \xi} - fv + \frac{g}{\sqrt{G_{\xi\xi}}}\frac{\partial \zeta}{\partial \xi} = -\frac{1}{P_0\sqrt{G_{\xi\xi}}}\frac{\partial P_{atm}}{\partial \xi} + F_\xi \tag{2}$$

$$\frac{\partial v}{\partial t} + \frac{u}{\sqrt{G_{\xi\xi}}}\frac{\partial v}{\partial \xi} + \frac{v}{\sqrt{G_{\eta\eta}}}\frac{\partial v}{\partial \eta} + \frac{uv}{\sqrt{G_{\xi\xi}}\sqrt{G_{\eta\eta}}}\frac{\partial \sqrt{G_{\xi\xi}}}{\partial \eta} - \frac{u^2}{\sqrt{G_{\xi\xi}\sqrt{G_{\eta\eta}}}}\frac{\partial \sqrt{G_{\xi\xi}}}{\partial \eta} + fu + \frac{g}{\sqrt{G_{\xi\xi}}}\frac{\partial \zeta}{\partial \xi} = -\frac{1}{P_0\sqrt{G_{\eta\eta}}}\frac{\partial P_{atm}}{\partial \eta} + F_\eta \tag{3}$$


where $\xi$, $\mathbf{\eta}$ are the spatial co-ordinates, $\zeta$ is representing water level above some horizontal plane of reference (m), u & v are the
velocities in the $\xi$ and $\mathbf{\eta}$ direction (m/s), d is the water depth below some horizontal plane of reference (m), g is the acceleration
of gravity (m/s$^2$), $P_{atm}$ is the atmospheric pressure at water surface ( kg/m/s$^2$), Q is the discharge of water, evaporation or
precipitation per unit area (m/s) , $\sqrt{G_{\xi\xi}}$ is the coefficient used to transfer one coordinate system into another one (m), $F_\xi$ are the
turbulent momentum flux in $\xi$-direction (m/s$^2$ ), $F_{\mathbf{\eta}}$   are the turbulent momentum flux in $\mathbf{\eta}$-direction (m/s$^2$ ). Along with the
appropriate set of initial and boundary conditions, the above-mentioned set of equations have been solved on an Arakawa-C type
finite difference grid. Delft3D- FLOW manual (Delft Hydraulics, 2006) contains detailed information about these numerical
aspects.



**2.1.2 Model Grid and Bathymetry**
The grid was set up using spherical coordinates, as displayed in Figure 1.  The grid spacing varies from a minimum of 125
meters to a maximum of 1140 meters. The finer resolution weas applied over land for calculating the inundation or wetting
process accurately. The bathymetries of the rivers and estuaries are specified using the cross sections measured by the Institute of
Water and Flood Management, Bangladesh. The land elevations are specified using the data from the Center for Environmental
and Geographic Information Services (CEGIS), Bangladesh. The ocean bathymetry is specified using the data from the General
Bathymetric Chart of the Oceans (BODC, 2003). Bathymetry and topographic data have been interpolated over the model
domain using triangular interpolation and grid-cell averaging methods.
**2.1.3 Wind and Pressure Field**
Track data of TCs Sidr and Aila were obtained from the Indian Meteorological Department ([www.imd.gov.in](www.imd.gov.in) ). Using those data
as input, TC surface winds and mean sea level pressure fields were generated using the Wind Enhancement Scheme (WES)
(Heming et al. 1995) method based on the analytical equation by Holland (1980). Delft3D slightly improved the original WES
by introducing TC asymmetry. Unlike some pervious method that incorporates TC wind asymmetry information from
observations (Xie et al. 2006), in WES the asymmetry was brought about by applying the translation speed of the cyclone center
displacement as steering current and by introducing rotation of wind speed due to friction (Heming et al. 1980).
According to the Holland's equation, gradient wind speed $V_g(r)$ at a distance r from the Centre of the cyclone is expressed as the
following:
$$V_g(r) = \left[ \frac{AB(p_n - p_c)\exp\left(-\frac{A}{r^B}\right)}{\rho r^B} + \frac{r^2 f^2}{4} \right]^{0.5} - rf/2 \qquad (4)$$
Here  $\rho$ is the density of air, $p_c$ is the central pressure and $p_n$ is the ambient pressure, the Coriolis parameter is represented by  $f$ .
A and B are determined empirically; with the physical meaning of A as the relation of pressure or wind profile relative to the
origin, and parameter B defining the shape of the profile. Delft3D introduces a central pressure drop of  $p_d = p_n - p_c$ . By
equating $\frac{dV_g}{dr} = 0$ , radius of maximum winds $R_w$ can be given as follows:
$$R_w = A^{1/B} \qquad (5)$$
Thus, $R_w$ is independent of the relative values of ambient and central pressure and is defined entirely by the scaling parameters A
and B. Substitutions lead to the expression for the maximum wind speed $V_m$
$$V_m = \left[ \frac{B p_d}{\rho e} \right]^{0.5} \qquad (6)$$
Complete details about this method can be found in the user manual of Delft3D Flow (Delft Hydraulics, 2006).
The circular grid of TC wind fields used in this study consists of 36 columns and 500 rows and the data were updated at 6 hourly
intervals throughout its movement until the landfall. Figure 2 shows a snapshot of the wind field of TC Sidr over the model
domain, before landfall, generated using Holland's equation above.





### 2.1.4 Roughness

The spatially varying Manning's Roughness value has been defined based on land cover, such as vegetation, rivers and ocean (Table 1). In the study domain, a mangrove forest, Sundarban, is located in the Southwest region, near TC Sidr's landfall location (Figure 1). Sakib et al. (2015) found that Sundarban plays a significant role as a buffer in reducing the total inundation during TC passages. Therefore, in this study, the mangrove region is considered.

In selecting the roughness values, methods described in Zhang et al. (2012) was followed and slightly modified values were defined for the study area based on the vegetation types in that area.

### 2.1.5 Boundary conditions

Upstream boundaries were specified as discharges at the mouths of the three major rivers; the Ganges, the Brahmaputra & the Upper Meghna; obtained from the Bangladesh Water Development Board (BWDB) as daily discharge. The downstream ocean boundary was defined by the Topex/Poseidon Inverse Tidal model, based on Egbert et al. (1994) Location of the downstream ocean boundary was shown in Figure 1

### 2.2 Calculation Procedure for Present Day and Future Storm Surge and Inundation Scenario

To generate storm surge and inundation for present day climate scenario, upstream discharge and downstream water level data from the present day were used. For future SLR scenarios, present day hydrodynamic conditions and the strengths of present day TCs were used but the future sea level was modified based on the SLR projections by Caesar et al. *(2017; under review)*. Scenarios were generated for both the Mid-21st century and the End-21st century time horizons for these TCs, Sidr and Aila. Finally, comparisons were made in terms of storm surge and inundation to identify the changes between present day and future SLR scenarios.

Now, future Storm surge inundation due to SLR is a probabilistic event that requires proper addressing of the uncertainties associated with the input parameters. To address the future tropical cyclone uncertainties and obtain statistically significant results, we created an ensemble of tropical cyclone tracks. The ensemble tracks were generated from different historical tropical cyclones that made landfall over the study domain with different intensities. Along with the uncertainties associated with future landfall locations, the intensity of Sidr-like and Aila-like TCs may be different. So, to address the uncertainty with the intensity, we increased and decreased their intensity by 10% to simulate a probable range of future storm surge inundation.

Storm surge inundation can also be different based on landfall timing. If the storm would make landfall during the high tide condition, flooding would be much higher at that time than what could happen during a low tide condition. We note that TC Sidr and TC Aila made landfall during the high tide conditions, which may not always be applicable for the future TCs. To also address uncertainties with the TC landfall timing, experiments were conducted by changing the timing of landfall to identify the impact of high tide and low tide on storm surge and inundation. Here in this study, future storm surge inundation scenarios caused by the ensemble tracks will then be simulated by incorporating the projected SLR. By taking all these parameters into consideration, we conducted a total 108 ensemble simulations (36 for each; present day and two SLR scenarios). Parameters that were considered in making ensemble projections shown in Table 2.





**3. Results**
**3.1 Validation of the Model**
Hourly tidal data from the Bangladesh Inland Water Transport Authority (BIWTA) was used to evaluate the performance of the
model used in this study. The model simulation's root mean square error (RMSE)[7], mean absolute error (MAE)[8] and dimension-
less Nash-Sutcliffe coefficient (E)[9] (Nash and Sutcliffe, 1970) were calculated and listed in Table 3. A Nash-Sutcliffe coefficient
ranges between (-ve)infinity (no skill simulation) and one (perfect simulation).

$RMSE = \sqrt{\frac{\sum_{i=1}^{n}(X_{obs,i}-X_{model,i})^2}{n}}$                                                                                                   (7)
$MAE = \frac{1}{n}\sum_{i=1}^{n}|X_{obs,i} - \hat{X}_{obs}|$                                                                                                   (8)
$E = 1 - \frac{\sum_{i=1}^{n}(X_{obs,i}-X_{model})^2}{\sum_{i=1}^{n}(X_{obs,i}-\overline{X_{obs}})^2}$                                                                                                   (9)

The simulated water levels were compared against the measured data from Bangladesh Inland Water Transport Authority
(BIWTA) at two locations: Barisal and Charchanga (Figure 1). Barisal station is located more towards the inland whereas
Charchanga is located near the coastline where the grid cell resolution was coarse. But none of them are in the open ocean water,
which is important to get a clear idea about storm surge level. TC Sidr made landfall near the Barisal Station (Figure 1) and the
impact of storm surge was clearer at the Barisal station than that of TC Aila, which made landfall outside the model domain
(Figure 1); therefore its impact was not as clear as that of Sidr.

In Figure 3(a) for TC Sidr at the Barisal station, the modeled water level, including storm surge and astronomical tides, was
slightly lower than the observations, and at the Charchanga station (Figure 3b) the measured water level variation displayed
larger amplitudes than did the model output, perhaps due to the coarse resolution of bathymetry. Similar types of variations
between measured and modeled water level was found for TC Aila (Figure 3c and Figure 3d). Nevertheless, the modeled water
level variations during TCs Sidr and Aila agreed reasonably well with measured data; as also confirmed by the average RMSE,
MAE and Nash-Sutcliffe coefficient. Therefore, we conclude that the method can be used to study the impact of SLR on storm
surge and inundation in future climate change scenarios.
**3.2 Present Day Inundation Scenario**

The storm surge inundation scenarios due to the two TCs considered were shown in Figure 4.
It can be seen from Figure 4 that the area flooded by TC Sidr  (yellow shade+red shade) was much higher than the area flooded
by TC Aila  (white shade+red shade), a result that is consistent with the fact that the category-5 TC Sidr was much stronger than
the category-1 TC Aila and directly hit the study area. The maximum sustained wind speed for TC Sidr was 260 km/h whereas
for TC Aila it was 110 km/h. The landfall location of Sidr was on the Eastern side of Sundarban, while for Aila, the landfall




location was towards the Western side of Sundarban. That explains why the inundations due to TC Sidr were located near the
eastern side of Sundarban, whereas for Aila, the inundation was located mainly in the western part. The extent of inundation due
to Sidr (1860 km$^2$) was 35% larger than that of Aila (1208 km$^2$)
Sakib et al. (2015) showed that Sundarban acted as a buffer zone in reducing the impact of Sidr and thereby reduced much of the
potential inundation depth and extent of flooding. As mentioned before, in the model simulation the impact of Sundarban was
realized using a higher Manning's roughness value as resistance for the surge to travel.
**3.3 Impact of Future Climate Scenarios on Storm Surge Inundation**

Future inundation scenarios were generated for two different time horizons: one for the mid-21st century and the other for the end
of the 21st century. The initial ocean water level was raised by 0.26 meters and 0.54 meters for the mid-21st century and end-21st
century, respectively. The upstream river discharge and downstream ocean water level were used from present day climate
scenarios.

In this section we seek to answer the question: if present day's TCs were to happen in future SLR scenarios, what storm surge
and inundation hazard would they cause? Therefore, the tracks and intensities of the two-present day TCs, Sidr and Aila, were
used as the model wind input parameters.

Figure 5 shows that under future SLR scenarios, the inundated areas would be significantly higher than those under the present-
day climate condition, as indicated by the white color shaded areas, for the TCs with the same strengths and landfall paths. For
the category-5 TC Sidr, the inundated area would be 31% and 53% higher than present day's 1860 km$^2$ inundated area, in mid-
21st century (0.26 meter SLR) and end of-21st century (0.54 meter SLR) climate scenarios, respectively (Figure 5a and Figure 5b)

Similarly, for TC Aila, a category 1 storm, there would be an increase in inundated areas. The calculated inundated area for TC
Aila under mid-21st century and end-of-21st century was found to be 1550 square kilometers and 1770 square kilometers
respectively (Figure 5c and Figure 5d) whereas for the present-day scenario it was found to be 1208 square kilometers.

All these simulations were done using present day tides, river discharges and the track and strength of present day TCs; while
changing the initial sea water level to reflect the effect of SLR. Therefore, the results suggest that even if the future TCs
strengths, the tides and river discharges remain the same as in the present, future SLR would significantly increase the inundated
area.
All of these simulations were done using present day tides, river discharges and the track and strength of present day TCs; while
changing the initial sea water level to reflect the effect of SLR. Therefore, the results suggest that even if the future TCs
strengths, the tides and river discharges remain the same as in the present, future SLR would significantly increase the inundated
area.
**3.3.1 Ensemble Projection of Future Storm Surge Inundation**
As discussed in section 2.2 that, future change in storm surge inundation can be different based on the intensity, landfall location
and timing of future TCs. By considering all those uncertainty factors mentioned in Table 2, a column plot was created (Figure



6) for present day sea level and future SLR scenarios. Ensemble simulation outputs also showed evidence for increase in the inundated area under the effect of SLR. For the present day scenario (black column) out of 36 simulations, frequency of storm surge inundation incidents that would likely occur between the range of 1000-1250 km$^2$ is 13 whereas for 0.26 meters of SLR(red column), peak of the column shifted towards right side with a maximum frequency of inundation events occurred within the range of 2000-2250 km$^2$ (10 times out of 36 simulation results). And for 0.54 meters of increase in sea level (blue column), the peak of the column shifted more towards the right and the maximum number of simulation outputs (11 out of 36 simulations) showed the range of inundation to be within 3500-3750 km$^2$. These results show that even the change in intensities of future TCs are indefinite and the landfall timing is uncertain, increase in sea level is going to increase the area of inundation.

**3.4 Impact of Sea Level Rise on Future Storm Surge Level**

In addition to the inundation area, SLR would also greatly affect storm surge levels. Similar to the approach used in the inundation study (Section 3.3), TCs Sidr- and Aila-induced storm surges in the future SLR scenarios were simulated using their recorded strengths.

The simulated storm surge water levels in future SLR scenarios were compared with both the observed and model generated ones under the present day scenarios (Figure 7). It is to be mentioned that, while generating the future water level under the effect of SLR, the baseline is only changing by considering the SLR effect and based on that factor the future storm surge level was calculated. Other than that, the water level is the same as present day TCs.

From Figure 7 we can see that, for the case of TC Sidr the simulated storm surge level would become 2.3 meters (Figure 7a) in Barisal station which is around 21% higher than the present day scenario. Similar to that, under the end-of-21st century 0.54 meters SLR scenario in Barisal, the storm surge would be around 37% higher (Figure 7a.) than the present day scenario and the peak water level would reach 2.6 m.

Increase in storm surge was found at the Charchanga station also. For TC Sidr, under the mid-21st century scenario (0.26 meter SLR), the model simulated storm surge level was found to be 14% higher (2.24 meters) (Figure 7b) than the present day and 31% higher (2.59 m) (Figure 7b) than the present day for the end-of-21st century (0.54 m SLR) climate scenario.

For TC Aila in Barisal, the Storm Surge would become 22% higher than the present day, which was 1.61 meters under the 0.26 meters SLR condition for the mid-21st century climate scenario (Figure 7c). During the end-of-21st century climate scenario, the increment would become even higher as the SLR would be 0.54 meter. Storm Surge would be 51% higher (1.96 meters) (Figure 7c) than the present day under the 0.54 meters SLR condition at the end of the century.

At Charchanga, the storm surge would be higher than the present day condition for TC Aila. For the mid-21st century under the 0.26 meters SLR scenario, storm surge would become 3.01 meters which is around 50% higher than the present day condition (Figure 7d). And for the end of 21st century, this would become 68% higher than the present day as the SLR would reach 0.54 m (Figure 7d).



The 0.26 m SLR for the mid-21$^{st}$ century would increase the water level, and the surge peak would be much higher at 2.4 m than the present day observed value at 2.0 m (Figure 7a). Figure 7b shows the same comparison, but for a 0.54 m SLR in the end-of-21$^{st}$ century scenario. For this case, the difference between the present day and end-of-21$^{st}$ century peak water level is much higher than what we found in the mid-21$^{st}$ century climate scenario.

**4. Discussions**

In this paper, we showed that even if the future TCs remain the same strength like the present day ones their impact will be much higher in a changing climate due to the effect of SLR. Several other factors not included in the modeling could make the storm surge and inundation situation far worse than that shown in the modeling result. These factors include mangrove coverage decrease, morphological changes, TC strength increase, and upstream river discharge changes.

For including the effect of future SLR in the model simulations, several methodologies were examined. One of the methods that we experimented in this study was to include the increased sea level in open ocean boundary instead of adding it in to the whole ocean depth by keeping the coastline fixed. In such case, an additional pressure gradient force was found acting towards the coast which made the inundated area much higher.

In order to make the future SLR simulation realistic, we considered the increased sea level in ocean bathymetry and increased the depth by 0.26 and 0.54 m, respectively, by considering land submergence near the coast. In that case, the result looked much realistic than the previous one and this is the method we followed in this paper. For example, for the case of TC Aila under the end-of-21st century scenario where we used a SLR of 0.54 m SLR at the open ocean boundary instead of adding it to ocean depth and using the hydrodynamic conditions from the present day, the total inundated area was found to be 79% higher than the present day one. Similar to that, for the mid-21st century scenario (a 0.26 m SLR), the inundated area was found to be 69% more than the present day scenario. But when we added the SLR in ocean depth, the mid-21$^{st}$ century and end-of 21$^{st}$ century inundated area was found to be 28% and 46% higher than the present day scenario. This increase in inundated area was much less than the one that we found by adding the SLR in the open ocean boundary. Figure 8 displays the differences in inundated area based on the consideration of SLR in the model input.

As discussed, TC Sidr made landfall near Sundarban, where the mangrove forest zone acted as a buffer in reducing the impact of the storm surge flood. That is why, even though it was a TC 5, its impact was not as high as it might have been expected to be. In this study, the roughness of the mangrove forest zone on the South-West part of Bangladesh was considered to be fixed for the present day as well as for future scenarios. But Mukhopadhyay et al. (2015) predicted that 17% of the total mangrove cover could disappear by 2105. If this decreasing trend of vegetation were considered in this study, the flooded area could be much higher.

Morphological changes were not considered in this study. But according to Goodbred et al. (2003), each year the eastern estuary, the central estuary and the western estuary are losing land at a rate of 0.13 cm/year, 0.16 cm/year and 0.16 cm/year, respectively. This could also lead to increased inundated areas for future scenarios. But as the focus of the paper is to predict the future scenario of storm surge and inundation due to the effect of SLR and comparisons with the present day scenarios, it is important that we keep the roughness and morphological changes constant so that consistent comparisons can be made.



Some previous research showed that there could be increases in hurricane strength and landfall probability in the future due to
global climate change (Haarsma et al. 2013, Bender et al. 2010, Bengtsson et al. 2007). Though we slightly modified the present-
day TC strengths and selected 12 historical TC tracks to reduce landfall uncertainties and to make ensemble projection of future
storm surge inundation, strength may be much higher than the ones that we considered for this study. In such case, the
devastation could well be much higher under projected SLR conditions; which is very alarming
In this paper, we used the present-day river discharge data as an upstream boundary for generating future inundation scenarios.
But using the INCA-N, an Inland Catchment Modeling system and considering projected climatic & socio-economic scenarios,
Whitehead et al. (2015) showed that, there will be a significant increase in future monsoon intensities due to the impact of
climate change. That would make future flooding scenarios much worse than those experienced presently. So, based on the
changes in TC intensity, river discharges and land-use changes, the situation could well become more badly impacted than what
we found in this study.
The findings of our study are important for local governments to consider while they make new management and policy
decisions and to improve TC preparedness plans by increasing numbers of shelters and heights. Generally, in TC shelters, the
first floor is kept transparent due to the risk of high surge waters. Our study showed that, in the future, there will be an increase
in surge level from a minimum of 15% up to 70% if a TC 1 or a TC 5 makes landfall under increased SLR conditions. So, the
authority may consider increasing the height of the first floor considering the future risk of increase in storm surge level and
safety of local populations. Also, our model outputs showed that, the inundated area increase would range from 28%-53%
percent if there's any TC 1 or TC 5 were to make landfall with SLRs of 0.26 m or 0.54 m. This shows that a huge number of new
areas are going to face the impacts of storm surge inundation and by considering this issue, it is high time to increase the number
of TC shelters in the coastal areas of Bangladesh.
**5. Conclusion**

Employing the Delft3D-FLOW model, we simulated coastal storm surge and inundation for present day and future SLR
scenarios and compared the changes between them. After validating the present day model, simulations were conducted for mid-
21$^{st}$ century and end-of-21$^{st}$ century climate scenarios where the SLR has been considered as 0.26 m and 0.54 m respectively.
The model results showed that, with an increase of 0.26 m and 0.54 m SLR, there would be an increase of 38% and 48% of
inundated area respectively if TC Sidr was to made landfall with its present day strength. There would also be an increase of
25% and 34% in inundated area if category-1 TC Aila would make landfall with its present day strength but under the condition
of 0.26 m and 0.54 m respectively. Outputs from the ensemble projections showed that, even if the TC intensities, landfall
location and timings are uncertain, the probable range of inundated are would shift from 1000-1250 km$^2$ (present day) to 2000-
2250 km$^2$ (0.26 meter SLR) and 3500-3750 km$^2$ (0.54 meter SLR). Besides the inundated area, we also investigated the changes
in storm surge level if TC Sidr and TC Aila would make landfall under future SLR conditions. Similar to the inundated area,
increases in storm surge levels were found for future scenarios. The significant increase in simulated storm surge and inundation
hazards highlights the need for the local governments to improve cyclone preparedness in future SLR scenarios




**Acknowledgement**
The authors would like to thank Coastal Carolina University's Cyber-infrastructure project (http://ci.coastal.edu) for providing
access to computational resources. Also, we would like to acknowledge Dr. Susan Kay from Plymouth Marine Laboratory, UK
for her thoughtful opinions regarding the SLR input in model and Institute of Water and Flood Management, Bangladesh for
providing important data & support for this work.

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






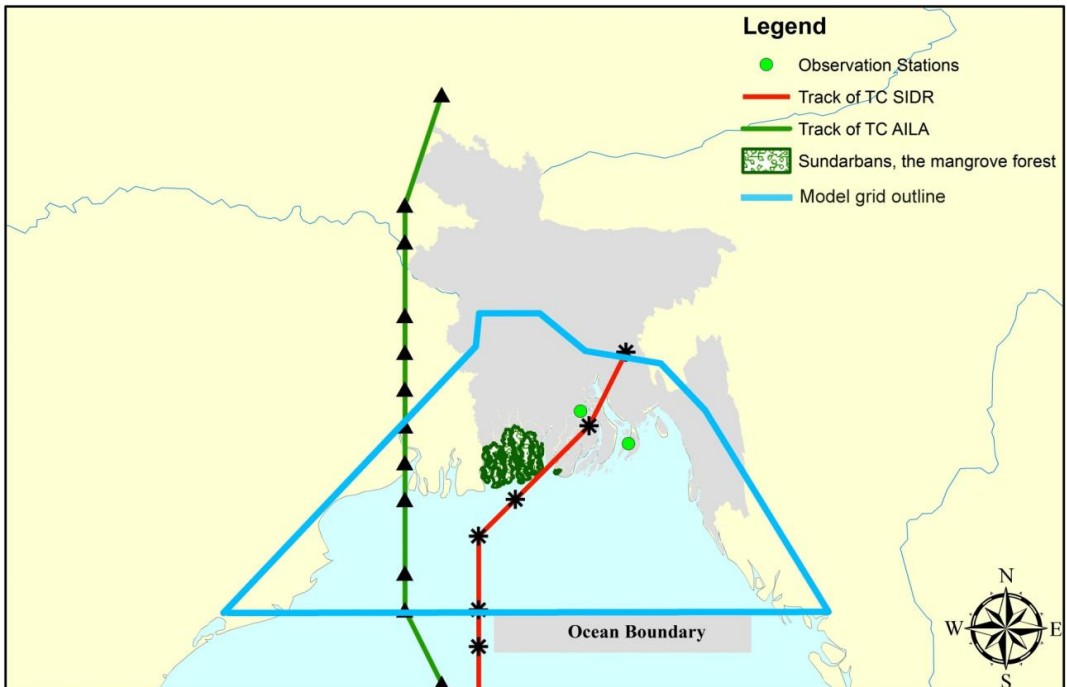

**Figure 1.** Map showing the study area for this work. The red and green lines representing the tracks of TC Sidr and TC Aila respectively. Area
marked with green color indicates the Sundarban mangrove forest region. Two green circles over the study area are the observation stations of
Bangladesh Inland Water Transport Authority (BIWTA). The blue colored outline shows the extent of model grid over the region.













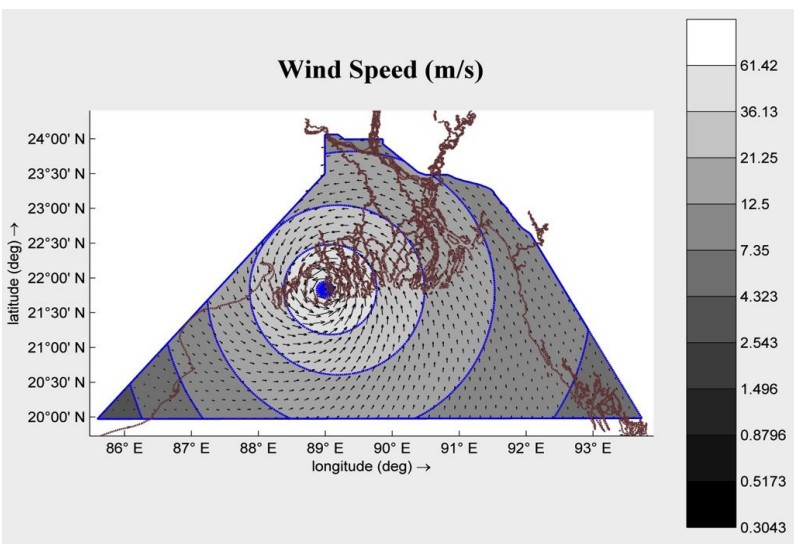

**Figure 2.** Distribution of wind field over the model domain for TC Sidr during landfall generated using Holland's Equation.
















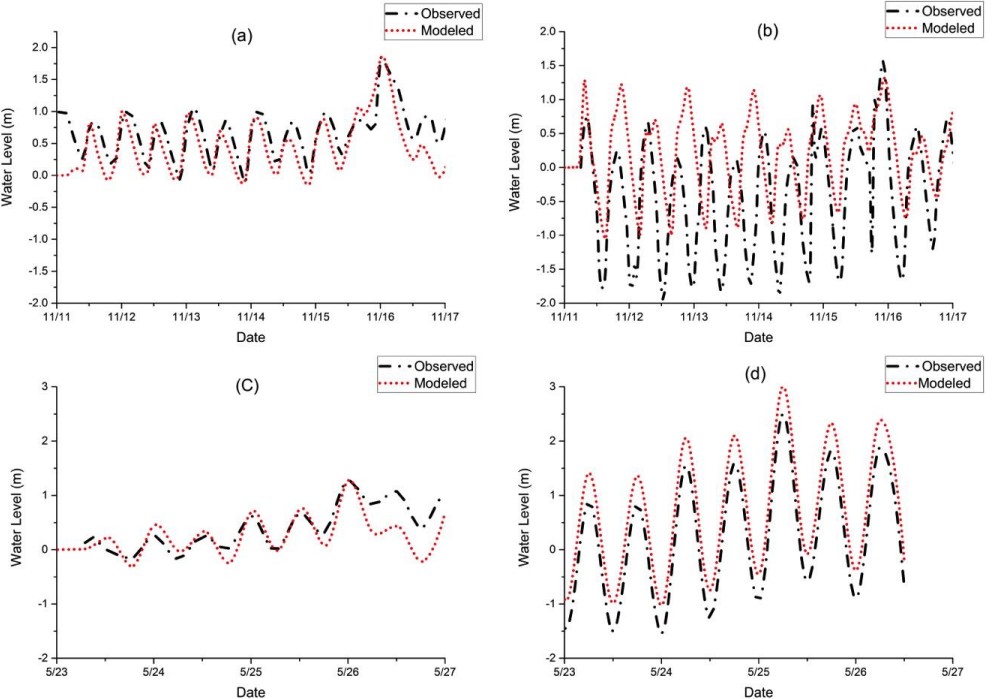

**Figure 3.** Comparison of observed and modeled Water Level for TC Sidr and TC Aila in Barisal and Charchanga observation stations. (a)
Measured and Modeled Water Level comparison for TC Sidr in Barisal, (b) Measured and Modeled Water Level comparison for TC Sidr in
Charchanga, (c) Measured and Modeled Water Level comparison for TC Aila in Barisal, (d) Measured and Modeled Water Level comparison
for TC Aila in Charchanga.










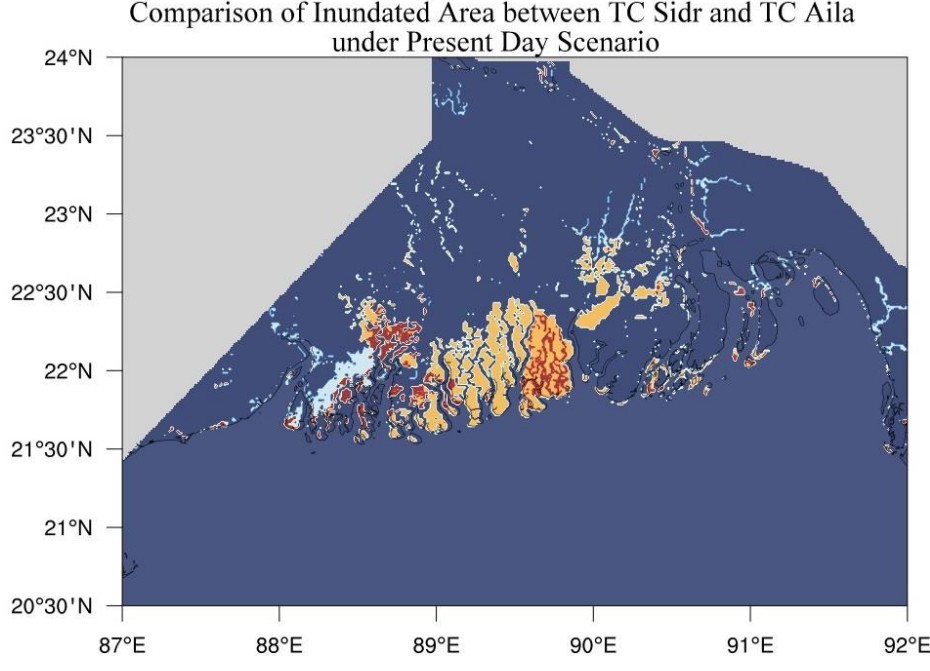


**Figure 4.** Yellow colors denotes the areas flooded by TC Sidr but not in Aila, and the white color representing the area inundated by TC Aila but not in Sidr. Red color is the area flooded by both TC Sidr and TC Aila. Blue color is showing the non-flooded area (either land or constant water).












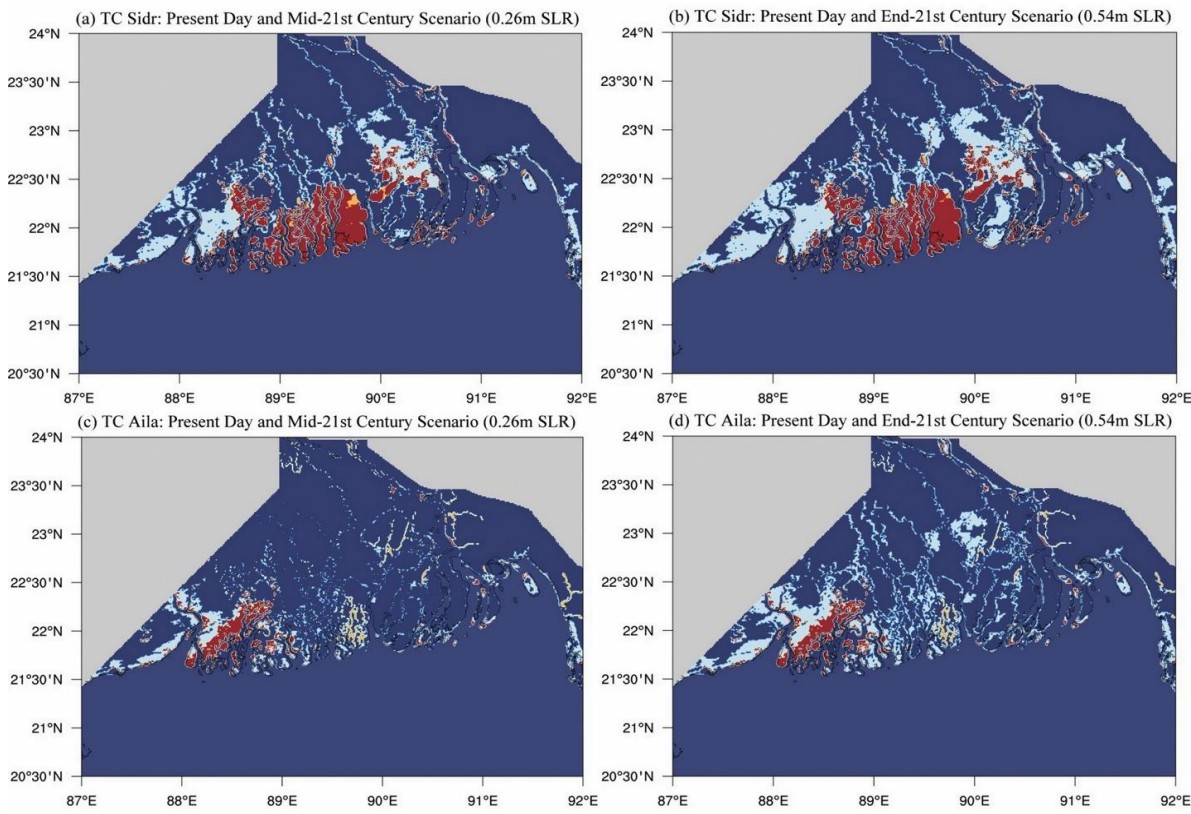

**Figure 5:** Comparison of inundated area between present day and future climate scenarios for (a) TC Sidr mid-21[st] century 0.26m SLR (b) TC Sidr end-21[st] century 0.54m SLR (c) TC Aila mid-21[st] century 0.26m SLR (d) TC Aila end-21[st] century 0.54m SLR. White color is representing the increased flooded areas that were not in present day scenario but the increase due to future SLR. Red color is showing the inundated areas that were similar both for present day and future SLR scenario case. Blue areas are non-flooded areas. Yellow color is showing the areas that were part of present day inundation but was not flooded during the future SLR conditions.













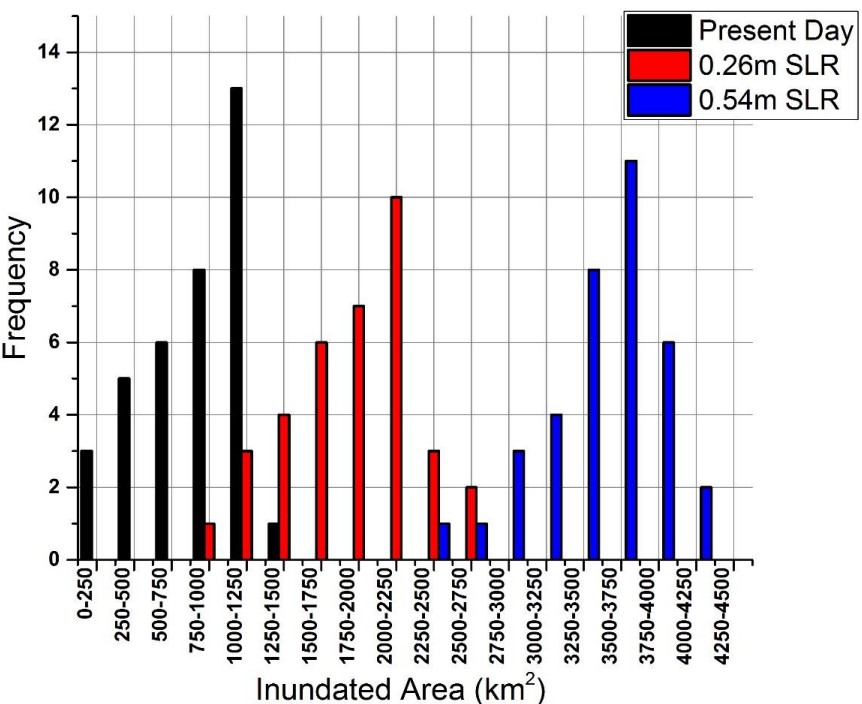

**Figure 6:** Ensemble projection of the future SLR impact on storm surge inundation. The column in black color is representing the inundation events for present day sea level condition, red colored one is for 0.26 meter of SLR and blue colored column is for 0.54 meter of SLR conditions. In total 108 simulations were conducted for present and two future SLR scenarios.






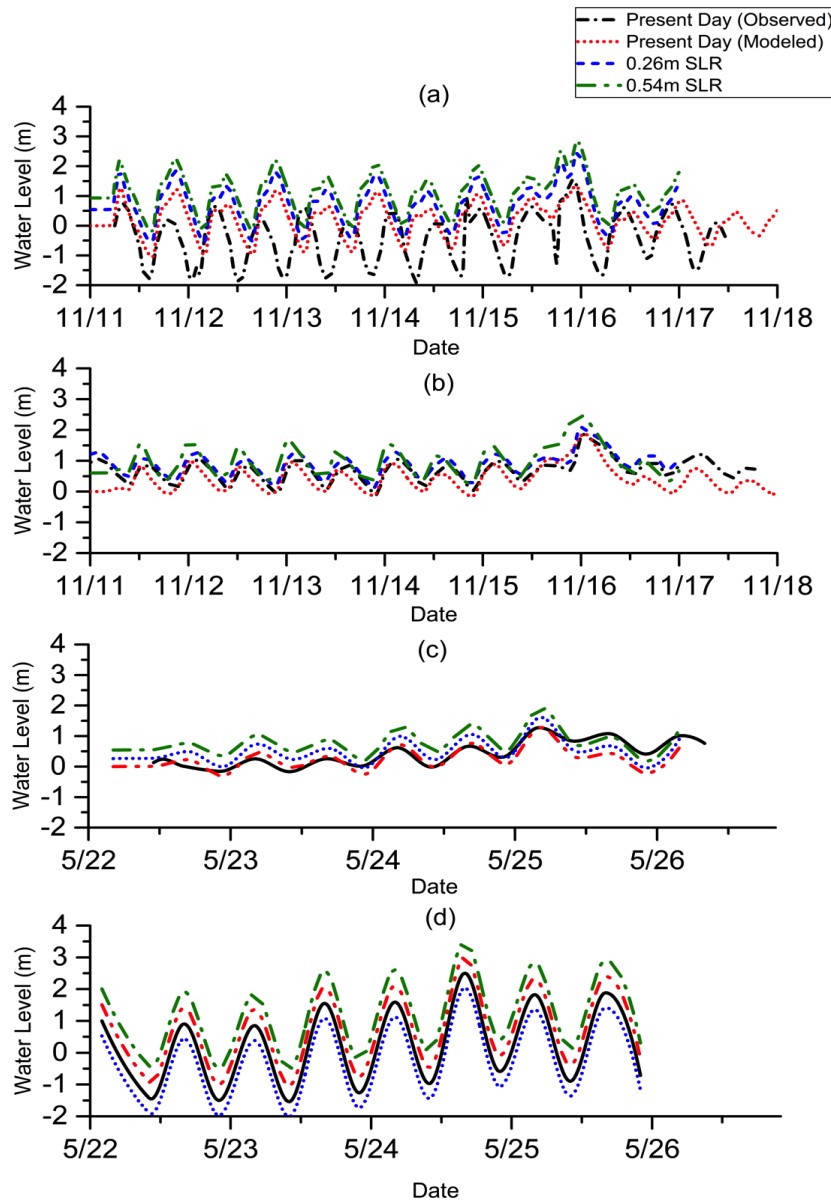

**Figure 7.** Comparison of water level for TC Sidr and TC Aila at Barisal and Charchanga Station between present day and future climate scenarios. (a) Comparison between present day and mid-21st century scenario (0.26m SLR) for TC Sidr in Barisal Station, (b) Comparison between present day and end-21st century scenario (0.54m SLR) for TC Sidr in Charchanga station, (c) Comparison between present day and mid-21st century scenario (0.26m SLR) for TC Aila in Barisal Station, (d) Comparison between present day and end-21st century scenario (0.54m SLR) for TC Aila in Charchanga station.







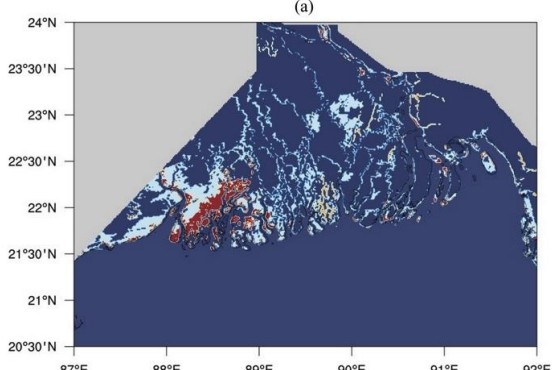 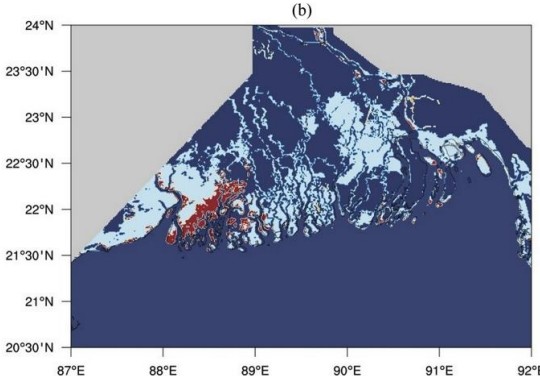


**Figure 8.** Comparison of inundated areas for TC Aila between present day and end-21$^{st}$ century (0.54m SLR) scenario. White color is representing the increased flooded areas that were not in present day scenario but the increase due to future SLR. Red color is showing the inundated area that's similar both for present day and future scenario case. Blue areas are either land or constant waters (those which are already water at the model initialization time). Figure (a) is representing the inundated area when SLR was considered on ocean depths instead of adding it in to the open ocean boundary and figure (b) showing the inundated area when we considered the SLR on ocean boundary.























**Table 1** Manning's Roughness Coefficient for different land coverings.

| Land cover | Manning's coefficient |
|---|---|
| River | 0.015 |
| Mangrove | 0.080 |
| Ocean | 0.01 |
| Land | 0.025 |




**Table 2:** Parameters considered for ensemble projection of storm surge inundation which includes the TC intensities, tidal conditions and the
SLR scenarios.

| TC name | Intensities | Tide conditions | SLR |
|---|---|---|---|
| TC Sidr | +10%, present day, -10% | High Tide, low tide, actual tide, zero tide | Present day, 0.26 meter, 0.54 meter |
| TC Aila | +10%, Present day, -10% | High Tide, low tide, actual tide, zero tide | Present day, 0.26 meter, 0.54 meter |
| 12 historical TC tracks | Actual intensities | Actual tide conditions | Present day, 0.26 meter, 0.54 meter |



**Table 3.** Computed values of RMSE, MAE and Nash-Sutcliffe coefficient for both TC Sidr and TC Aila

| Stations | TC Sidr | | | TC Aila | | |
|---|---|---|---|---|---|---|
| | RMSE (m) | MAE (m) | NASH | RMSE (m) | MAE (m) | NASH |
| Barisal | 0.23 | 0.16 | 0.85 | 0.33 | 0.24 | 0.65 |
| Charchanga | 0.26 | 0.19 | 0.80 | 0.28 | 0.17 | 0.73 |
| **Average** | **0.245** | **0.175** | **0.825** | **0.305** | **0.205** | **0.69** |