# Peer review of "Ensemble Projection of the Sea Level Rise Impact on Storm Surge"

_Natural Hazards and Earth System Sciences, 2017_

## Referee Comment (RC1) · Anonymous Referee #1 · 10 Aug 2017

This paper presents the results of numerical experiments to investigate the impact of sea level rise induced by climate change on the extent and severity of the inundation caused by tropical cyclone (TC) storm surges in coastal Bangladesh. As the authors correctly point out, the approach used is a quite simple one and does not take into account relevant aspects such as possible future modifications in TC tracks and intensity and morphological changes. Nevertheless, the manuscript is an original contribution to the issue of climate change-induced hazards and the results of the study provide interesting suggestions for mitigation measures to be taken by policy makers and can encourage further research about this topic.

[Figure]

The manuscript is quite well organized and written, even though the presentation of the results is somewhat confusing in some parts and should be revised and made clearer. In my opinion, the paper can be accepted for publication after the following comments are addressed.

General comment: The description of results in Sections 3.2-3.4 can be sometimes confusing and has to be improved. The authors in some cases comment about absolute values of inundated areas extent in km2 or storm surge height, in other cases provide information about percent variations of simulated values with respect to present time and percentages related to the same quantities appear even inconsistent (see, e.g. lines 363 and 372). This is particularly the case of Section 3.4, where the discussion about Figure 7 is hard to follow. Lines 310-314 seem to repeat what stated in lines 295-301 but numbers are slightly different. The caption of Figure 7 itself is wrong, because each of the four plots shows the comparison of present time water level with both the considered future scenarios. Also, the higher percent variation in storm surge height at Charchanga station obtained for TC Aila with respect to Sidr is not intuitive and the authors should provide some interpretation attempt. In conclusion, my suggestion is to thoroughly revise this sections and to add one or more tables (e.g. one for inundation extent and another one for storm surge height) containing both the absolute values and the percent variations with respect to the present time scenario.

Specific comments:

Please check the correspondence between the references in the list and the citations in the text. For instance, the works by Mohal et al. (2006) and by Vatvani et al. (2002) seem to be missing in the text. Also, the reference to the Delft3D-FLOW manual is not coincident (Delft Hydraulics in the text vs. Hydraulics, D. in the reference list.

A general revision of the whole text is needed to eliminate several typing and punctuation errors, uppercase and lowercase letter usage, and missing or unnecessary blanks. For istance, see lines 73, 74, 136, 154, 181, 184, 190, 193, 240, 241, 279, 297, 351,

380.

Line 55: replace locale with locales.

Line 310-314 are redundant.

At the end of the Introduction, a brief paragraph illustrating the structure of the manuscript should be added.

In Equations (2) and (3), the term P0 is not defined.

Line 143: replace weas with was.

Lines 145-147: please provide some information about the native resolution of the topography and bathymetry data used.

Line 156: the reference should be to the work by Holland (1980), I suppose.

In Equation (6) the term e is not defined.

In Equation (8) the definition of MAE is not correct.

Line 218-219: the BIWTA acronym has been already introduced and can be used without the full explanation.

Line 376: replace "the probable range of inundated are" with "the most probable range of inundated area extent".

Line 480: replace representing with represent.

Lines 506-508: uppercase letter are unnecessary for measured and modeled water level.

Line 556: replace showing with is showing.

Table 2: 12 historical TC tracks are used in ensemble projection as mentioned in Sections 2.2 and 3.3.1. In my opinion, a further figure illustrating each track and/or just a table listing the main characteristics of each storm (e.g. name, intensity, day of landfall,

etc) would be useful.

Table 3: the third row with average values of statistical indicators can be eliminated, because averaging just two values is poorly significant.
[Figure]

---

## Referee Comment (RC2) · Anonymous Referee #2 · 15 Aug 2017

The paper addresses the potential influence of SLR on the flooding exposure during tropical cyclones in Bangladesh. The hydrodynamic model Delft-FLOW is used to simulate two historical storms and some future scenarios reflecting SLR of different magnitudes and uncertainties due to TC intensity and timing with respect to the tidal phase. Model results are validated against tide gauge measurements during the TC events. Further the authors investigate how inundated area and storm surge height would be changed if SLR would be present, assuming the properties of TCs remain unchanged. They conclude that even considering uncertainties of present-day TC properties the amount of flooded land would increase dramatically.

[Figure]

The paper is well composed and clearly written. The objectives, methodology and results are sufficiently described. The discussed problematic is relevant and adds to the insight of the SLR consequences at regional scale. The approach of separating the sources of uncertainty and selecting a single trigger (here SLR) for the analysis contributes to better understanding of the potential changes in the local system. However, as the authors also pointed out, there are many unconsidered conditions like changes in TCs intensity, morphology, river discharge, etc. This undermines the value of particular resultant numbers if they are considered without more generalized conclusions. For example, what would happen if the SLR of 0.4 or 0.7m occurs? Some discussion going beyond the presented two SLR case studies would be appropriate. I propose to put the results described in the Section 3.4 in relation with the SLR values and not only with the present-day inundated area/storm surge height. It would be interesting to see the direct comparison (in % or absolute values) of changes in storm surge height with respect to SLR, e.g. for TC Sidr and Charchanda it would be 0.27m increase for 0.26m SLR, which is basically a linear addition of SLR magnitude on top of the present-day surge, and 0.62m change for 0.54m of SLR, which has a considerable non-linear contribution. This could give an insight into the (non)linearity of the storm surge and SLR interactions for particular area.

It would be very helpful to include the terrain map of the model region with land elevation and land/water mask for better understanding of the present day situation and possible impacts. It could be combined with Figure 1 or not. Some names on Figure 1 would be also helpful (like main rivers, measurement location names, etc).

Minor comments:

- p2. lines 55-56: ". . .causing deaths . . . of lives". Please reformulate ('causing deaths of people' OR 'causing loss of lives')

- p3. line 83: please coordinate singular/plural forms "the impact . . . are debatable"

- p4. lines 110-114: this passage looks like repetition of the previous one. Please

remove or reformulate.

- p5. line 143: "... weas ..." typo?

- p5. lines 151-171: It is not quite clear from this description whether Delft3D has special module for generating wind and pressure fields from the TC track. If yes, and it "... slightly improves the original WES...", why the authors are still using WES and not Delft3D? If not, and wind and pressure are firstly generated by WES method and then fed to the Delft3d, then the description is misleading.

- p6. line 193: "... Storm surge..." -> "storm surge"

- p7. line 209: by tidal data the water elevation is meant? What type of instrument has been used, tide gauge?

- p7. line 212: "(-ve)infinity ..." typo?

- p8. lines 269-272: repeated passage, please remove

- p10. lines 310-313: please review or remove the passage, it does not describe Figure 7.

- p10. line 317: "... future TCs remain the same strength..." – either 'keep the same strength' or 'remain of the same strength'

- p10. line 322: "one of the methods we experimented in this study..." – either 'methods with which we experimented...' or 'methods that we tested'

- p10. line 327-328: "... the results looked much realistic..." -> ... the results looked much more realistic...

- p10. Lines 344-345: "...focus of the paper is to predict the future scenarios..." – please change 'predict' to assess/estimate/develop "... and comparison with..." -> 'and to compare with'

- p11. line 360: "... first floor is kept transparent..." – What does transparent mean in

this context? Why is it relevant here?

- Firgure 3b and 7a: for the case of TC Sidr in Charchanga the timeseries of measured and modelled water levels look somewhat different and out of phase. Do the authors have an explanation for this?

---

## Referee Comment (RC3) · Anonymous Referee #3 · 31 Aug 2017

The paper uses numerical modelling to analyze the effects of sea level rise (SLR) on the storm surge generated by tropical cyclones (TC) in the Bangladesh coast and the associated inundation on that area. Model results are validated using observations of two previous TC and a number of additional simulations are made to study future scenarios. The manuscript is pretty well written, although it can be improved following the suggestions detailed below. Besides the assumptions made to simplify the high level of uncertainty, the obtained results show how SLR would increase the inundation associated to TC in this area and can help coastal managers to design adaptation measures to deal with these problems. Therefore, the manuscript fits the scope of NHESS and may be published provided the authors address the following comments.

[Figure]

General comments

- The authors should justify why they use the SLR projections from AR4 (IPCC, 2007) (line 60) instead of those from AR5 (IPCC, 2013), although they are based on the worst AR4 scenario (A1F1, line 72). Taking into account that regional SLR rates are much higher than the global rate (lines 64-66) and that global SLR projections from AR5 are worse than AR4, the scenarios considered by the authors could be too optimistic.

- A number of geographical sites are cited in the text (e.g. Bay of Bengal and Andaman Sea (line 37); Ganges, Brahmaputra, Meghna rivers (line 92); Baguna (lines 97 and 100); Patuakhali (lines 99 and 106); Khulna (lines 100 and 106); Jhalokati (line 100); Chandpur (line (106), Sundarban (lines 238, 239, 240)) that should be placed in a map to facilitate the reading of the text. In the same way, a figure showing the topography of the area would be very useful. In addition, the shorelines should be clearer in figures 1, 4, 5 and 8, to better understand the magnitude of the flooded areas.

- In lines 201-206 the authors discuss the potential influence of the tide level on the inundation and indicate that different simulations have been performed considering diverse tide conditions, which are summarized in Table 2. However, nowhere is the magnitude of the tides shown. A description of tide features is necessary to understand the influence of this factor in the inundation.

- The writing of sections 3.2, 3.3, 3.4 and 4 is a little bit confusing with the mixing of percentages, inundation areas and water levels. Perhaps the results could be summarized in a table to ease the understanding of the changes associated to each scenario.

- Lines 321-324: In the discussion about the used methods, the authors say that they "included the increased sea level in open ocean boundary instead of adding it into the whole ocean depth". In my opinion this makes no sense because it introduces a discontinuity in the water level that physically is not possible. As the authors say, this produces an additional pressure gradient force acting towards the coast. Therefore, the obtained results are spurious. I suggest removing any reference to this method,

including figure 8.

- Some of the presented results seem inconsistent:

o In Figure 5 the comparison of inundated areas between present day and future climate scenarios is shown. In this figure, there are several small areas of yellow color indicating zones flooded under present conditions but not flooded during future SLR conditions. The authors should explain why these low lying coasts are flooded with present SLR and not with higher SLR, contrary to what would be expected.

o In lines 226-227 the authors say: "the measured water level variation displayed larger amplitudes than did the model output". Observing Figure 3b, the trend seems the opposite (for positive values) and the red line (modeled) is located above the black one (observed). On the contrary, negative values and total oscillations are greater in the case of observed data. I suggest clarifying this point.

o When comparing water levels of Figure 7 and Figure 3, the observed and modelled values are different in panels (a) and (b) of both figures. It looks like in one of both figures, these panels are exchanged.

Specific comments

- Lines 55-56: "the deaths of hundreds of thousands of lives". Better "the loss of hundreds of thousands of lives". This sentence is very similar to the following one: "This type of coastal flooding...", so probably both sentences could be combined into one.

- Line 83: "The impact of climate change..... are still debatable" should be "The impact of climate change..... is still debatable" or "The impacts of climate change..... are still debatable".

- Line 88: "will be method of this study", better "will be the method of this study".

- The name of a district is written differently: Patuakhali (line 99), Patukhali (line 106),

[Figure]

Pataukhali (line 106). Please be consistent and use only one name.

- Lines 110-114: This paragraph seems a repetition of a previous one.

- Lines 129-130: P0 and f are not defined in equations (2) and (3).

- Line 156: The reference Heming et al. (1980) is missing or there is a mistake and should be Heming et al. (1995).

- Line 167: The meaning of e is not defined in equation (6).

- Line 178: "methods described in Zhang et al. (2012) was followed" should be "methods described in Zhang et al. (2012) were followed".

- Line 184: "boundary was shown in Figure 1", better "boundary is shown in Figure 1". - Line 206: "...in making ensemble projections shown in Table 2" should be "...in making ensemble projections are shown in Table 2".

- Line 212: "(-ve)" looks a typo.

- Line 215: Equation (8) is wrong. The MAE is obtained by comparing observations with model results.

- Line 234: "the two TCs considered were shown in Figure 4.", better : "the two TCs considered are shown in Figure 4.".

- Lines 262-263: Please substitute "square kilometers" by "km2".

- Lines 269-272: This paragraph is a repetition of the previous one.

- Lines 310-313 are redundant with the previous paragraphs and although they coincide with Figure 7 caption (which is wrong), they do not describe Figure 7.

- Line 351: "SLR conditions; which is...", better "SLR conditions, which is...".

- References: Alam (1996), Mohal et al. (2006) and Vatvani et al. (2002) are listed in References but are not cited in the text.

[Figure]

- The reference corresponding to Delft3D model is cited in the text as Delft Hydraulics (2006) but is listed as Hydraulics, D. (2006). Please be consistent.

- Figure 7 caption is wrong and it does not describe this figure, since the results of both future scenarios are included in each figure.

---

## Author Comment (AC1) · 24 Sep 2017

**Reply to Anonymous Referee #1**

*This paper presents the results of numerical experiments to investigate the impact of sea level rise induced by climate change on the extent and severity of the inundation caused by tropical cyclone (TC) storm surges in coastal Bangladesh. As the authors correctly point out, the approach used is a quite simple one and does not take into account relevant aspects such as possible future modifications in TC tracks and intensity and morphological changes. Nevertheless, the manuscript is an original contribution to the issue of climate change-induced hazards and the results of the study provide interesting suggestions for mitigation measures to be taken by policy makers and can encourage further research about this topic. The manuscript is quite well organized and written, even though the presentation of the results is somewhat confusing in some parts and should be revised and made clearer. In my opinion, the paper can be accepted for publication after the following comments are addressed.*

We appreciate the comments from referee and would like to thank for evaluations and feedback which helped to improve the manuscript.

***General comment #1:*** *The description of results in Sections 3.2-3.4 can be sometimes confusing and has to be improved. The authors in some cases comment about absolute values of inundated areas extent in km2 or storm surge height, in other cases provide information about percent variations of simulated values with respect to present time and percentages related to the same quantities appear even inconsistent (see, e.g. lines 363 and 372). This is particularly the case of Section 3.4, where the discussion about Figure 7 is hard to follow. Lines 310-314 seem to repeat what stated in lines 295-301 but numbers are slightly different. The caption of Figure 7 itself is wrong, because each of the four plots shows the comparison of present time water level with both the considered future scenarios. Also, the higher percent variation in storm surge height at Charchanga station obtained for TC Aila with respect to Sidr is not intuitive and the authors should provide some interpretation attempt. In conclusion, my suggestion is to thoroughly revise this sections and to add one or more tables (e.g. one for inundation extent and another one for storm surge height) containing both the absolute values and the percent variations with respect to the present time scenario.*

Thank you for pointing out these problems. In the revised manuscript, we've updated it by adding both the absolute values and percent variations in the write up to make it easier to follow. Three separate tables were also added both for inundated area and storm surge level and their percent change to identify the differences easily.

In section 3.4, corrections were made in calculated values of storm surge level in following lines:

Line 295 [Revised line 287]: 2.13 meters instead of 2.3 meters.

Line 296 [Revised line 288]: 13.7% instead of 21%.

Line 297 [Revised line 289]: 28.67% instead of 37%.

Line 298 [Revised line 290]: 2.41 m instead of 2.6 m.

Line 300 [Revised line 292]: 13.95% instead of 14% …. 1.87 meters instead of 2.24 meters……33.45% instead of 31%

Line 301 [Revised line 293]: 2.19 m instead of 2.59 m.

Line 302 [Revised line 296]: 21.93% instead of 22%..........1.299 meters instead of 1.61 meters.

Line 304 [Revised line 298]: 50.96% instead of 51%

Line 307 [Revised line 301]: 3.075 meters instead of 3.01 meters…….23% instead of 50%

Line 308 [Revised line 302]: 55% instead of 68%.

A new table with all the calculations was also added in the manuscript.

Based on the corrected calculations, we've also updated figure 7. In the initial submission, Figure 7a was mentioned as "TC Sidr at Barisal" and Figure 7b was mentioned as "TC Sidr at Charchanga". Actually, Figure 7a was representing TC Sidr at Charchanga and Figure 7b was representing TC Sidr at Barisal. We've corrected these mistakes in the updated manuscript.

We've also corrected the calculation error in line 363 [revised line 364] and 372 [revised line 373]. In line 363 [revised line 364], it should be 28.3% - 53% as shown in newly added Table 5. In line 372 [revised line 373], it was incorrectly written as 38% and 48%. It should be 31% and 53%, based on the calculation shown in Table 5. In the revised version, it was corrected. Also, in line 374 [revised line 374], the percentage values were corrected and it should be 28.3% and 46.5% instead of 25% and 34%.

Caption of Figure 7 was also updated based on the suggestion.

An additional paragraph and a new Figure (Figure 8) was added based on the comment of reviewer #2 to represent the relation between SLR and the additional increase of storm surge level. Following underlined paragraph was added at the end of section 3.4 [Revised line 304-314]:

To analyze the linearity/non-linearity of storm surge level with respect to SLR, we conducted additional experiments based on 5 SLR scenarios; present daypresent-day sea level, 0.26 m of SLR, 0.33 m of SLR, 0.4 m of SLR, 0.47 m of SLR, 0.54 m of SLR, respectively. Results from these experiments are presented in the Figure 8.

For the case of TC Sidr in Barisal and Charchanga station, storm surge level increased almost linearly with respect to the addition of water due to the effect of SLR. For example, with a SLR of 0.47 m, the increase of storm surge level with respect to present day in Barisal and Charchanga stations were 0.453 m and 0.445 m, respectively (Figure 8a). On the other hand for the case of TC

Aila, with a SLR of 0.26 m, the increase in storm surge level were found 0.285 m and 0.575 m respectively for the Barisal and Charchanga station (Figure 8b). Though the storm surge level is increasing almost linearly with the addition of sea water, however, there's are still differences found between them. This could be influenced by the modification of ocean bathymetry to incorporate the effect of SLR. The margin of differences is higher for the Charchanga station comparing it with the Barisal station. The coarse resolution of topography near that area might be responsible for that.

Following are the four tables that were added in the updated manuscript for section 3.3 and section 3.4, updated Figure 7 and newly added Figure 8

**Table 5.** Comparison of inundated area between present day & future SLR scenarios and calculated change in percentage with respect to present day scenario.

| Scenario | TC Sidr | | TC Aila | |
|---|---|---|---|---|
| | Inundated Area | (%) change | Inundated Area | (%) change |
| Present Day | 1860 | | 1208 | |
| Mid-century | 2436.6 | +31 | 1550 | +28.3 |
| End-century | 2845.8 | +53 | 1770 | +46.5 |

**Table 6.** Comparison of storm surge level between present day and future SLR scenarios for the case of TC Sidr

| Scenario | Barisal | | Charchanga | |
|---|---|---|---|---|
| | Storm surge level (m) | % increase | Storm surge level (m) | % increase |
| Present Day | 1.873 | | 1.641 | |
| Mid-century (0.26m) | 2.13 | 13.72 | 1.870 | 13.95 |
| End-century (0.54m) | 2.41 | 28.67 | 2.19 | 33.45 |

**Table 7.** Comparison of storm surge level between present day and future SLR scenarios for the case of TC Aila

| Scenario | Barisal | | Charchanga | |
|---|---|---|---|---|
| | Storm surge level (m) | % increase | Storm surge level (m) | % increase |
| Present Day | 1.299 | | 2.5 | |

| | | | |
|---|---|---|---|
| Mid-century (0.26m) | 1.584 | 21.93 | 3.075 | 23 |
| End-century (0.54m) | 1.961 | 50.96 | 3.875 | 55 |

[Figure]

Figure 7. Comparison of storm surge water level between present day and future SLR scenarios. (a) TC Sidr at Barisal (b) TC Sidr at Charchanga (c) TC Aila at Barisal (d) TC Aila at Charchanga. The observed, modeled present-day, mid-of-21st century and end-of-21st century storm surge levels are denoted by the solid, red dashed, blue dotted, and greenred dash-dottted lines, respectively.

[Figure]

**Figure 8.** Relation between SLR and increase in storm surge level with respect to the present-day simulated water level for TC Sidr and TC Aila. (a) is representing the relation for TC Sidr and (b) is representing the relation for TC Aila..

**Specific comments:**

**#1**

*Please check the correspondence between the references in the list and the citations in the text. For instance, the works by Mohal et al. (2006) and by Vatvani et al. (2002) seem to be missing in the text. Also, the reference to the Delft3D-FLOW manual is not coincident (Delft Hydraulics in the text vs. Hydraulics, D. in the reference list.*

Mohal et al. (2006) was removed from the reference list since it was not cited in the main text. Vatvani et al. (2002) should be in section 2.1.3 but somehow it went missing. In the updated manuscript, it was added at the end of section 2.1.3. Along with that, we've also added the refence from Holland (1980) at the end of section 2.1.3.

Reference to the Delft3D-FLOW manual was also corrected in the reference list. Thank you for pointing out these errors.

*A general revision of the whole text is needed to eliminate several typing and punctuation errors, uppercase and lowercase letter usage, and missing or unnecessary blanks.*

A general revision has been carried out to eliminate typing, punctuation and grammar errors.

*#2 Line 55: replace locale with locales.*

Corrected

*#3 Line 310-314 are redundant.*

Removed.

*#4 At the end of the Introduction, a brief paragraph illustrating the structure of the manuscript should be added.*

Thank you for the suggestion. We've added the following paragraph at the end of introduction section: [Revised line 106-110]:

The structure of the paper is as follows: brief description of the Delft3D Flow model and the methodologies used to simulate future changes in storm surge and inundation, to generate ensemble projections of storm surge inundation were discussed in section 2, In section 3, validation of the model results, present day storm surge inundation scenarios, ensemble projection of storm surge inundation and future change in storm surge level were presented. Section 5 includes, discussion on model results and the uncertainties associated with the future projections. Finally, section 5 presents the concluding remarks on research findings.

*#5 In Equations (2) and (3), the term P0 is not defined.*

It's the density of water. We've corrected it in the updated version.

*#6 Line 143: replace weas with was.*

Corrected.

*#7 Lines 145-147: please provide some information about the native resolution of the topography and bathymetry data used.*

We've added the details on bathymetry and topography section.

In line 145, "The land elevations are specified using the data from the Center for Environmental and Geographic Information Services (CEGIS), Bangladesh" since they're based NASA's Shuttle Radar Topography Mission (SRTM) 90m resolution datasets.

Following lines were added in the 2.1.2 Model Grid and Bathymetry sections:

In this study, the land topography data were obtained from NASA's Shuttle Radar Topography Mission (SRTM) 90-m resolution datasets (Figure 1b). The ocean bathymetry wais specified using the data from the General Bathymetric Chart of the Oceans 30-arc-sec interval gridded data (BODC, 2003, Figure 1b).

*#8 Line 156: the reference should be to the work by Holland (1980), I suppose.*

Actually, it should be Hemming et al. (1995) instead of Hemming et al. (1980). Delft3D uses Wind Enhancement Scheme (WES) which is based on Holland's Wind Model (1980) in order to bring asymmetry by applying the translation speed of the cyclone center displacement as steering current and by introducing rotation of wind speed due to friction.

*#9 In Equation (6) the term e is not defined.*

It's the base of the natural logarithm (=2.71828182846) (Delft Hydraulics, 2011). We've included this information in the revised version.

**10 In Equation (8) the definition of MAE is not correct.**

*Corrected.*

*#11 Line 218-219: the BIWTA acronym has been already introduced and can be used without the full explanation.*

Corrected.

*#12 Line 376: replace "the probable range of inundated are" with "the most probable range of inundated area extent".*

Replaced.

*#13 Line 480: replace representing with represent.*

Replaced.

*#14 Lines 506-508: uppercase letter are unnecessary for measured and modeled water level.*

Corrected.

*#15 Line 556: replace showing with is showing.*

Corrected.

*#16 Table 2: 12 historical TC tracks are used in ensemble projection as mentioned in Sections 2.2 and 3.3.1. In my opinion, a further figure illustrating each track and/or just a table listing the main characteristics of each storm (e.g. name, intensity, day of landfall, etc) would be useful.*

Thank you for the suggestion. We've included Table 2 with the information of 12 historical TC tracks that were used for ensemble projections.

**Table 2** List of 12 historical TC events used for ensemble projection of storm surge inundation

| Name | Date | Landfall |
|------|------|----------|
| Tropical storm 13 | 14-18 November, 1973 | Noakhali |
| Cyclone 12 | 23-28 November, 1974 | Bhola |
| Tropical storm 19 | 07-12 November, 1975 | Chittagong |
| Tropical storm 1 | 22-25 May, 1985 | Noakhali |
| Cyclone 4 | 21-30 November, 1988 | Khulna |
| Cyclone 2 | 22-30 April, 1991 | Chittagong |
| Cyclone 2 | 26 April – 30 May, 1994 | Cox's Bazar |
| Cyclone 4 | 18-25 November, 1995 | Cox's Bazar |
| Cyclone 1 | 13-20 May, 1997 | Noakhali |
| Tropical storm 4 | 24-27 October, 2008 | Barguna |
| Tropical storm Mahasen | 10-16 May, 2013 | Patuakhali |
| Tropical storm Roanu | 18-21 May, 2016 | Chittagong |

*#17 Table 3: the third row with average values of statistical indicators can be eliminated, because averaging just two values is poorly significant.*

Thank you. It was eliminated.

---

## Author Comment (AC2) · 24 Sep 2017

**Reply to Anonymous Referee #2**

*The paper addresses the potential influence of SLR on the flooding exposure during tropical cyclones in Bangladesh. The hydrodynamic model Delft-FLOW is used to simulate two historical storms and some future scenarios reflecting SLR of different magnitudes and uncertainties due to TC intensity and timing with respect to the tidal phase. Model results are validated against tide gauge measurements during the TC events. Further the authors investigate how inundated area and storm surge height would be changed if SLR would be present, assuming the properties of TCs remain unchanged. They conclude that even considering uncertainties of present-day TC properties the amount of flooded land would increase dramatically. The paper is well composed and clearly written. The objectives, methodology and results are sufficiently described. The discussed problematic is relevant and adds to the insight of the SLR consequences at regional scale. The approach of separating the sources of uncertainty and selecting a single trigger (here SLR) for the analysis contributes to better understanding of the potential changes in the local system.*

We would like to thank the referee for the evaluation and for providing feedback which helped to improve the manuscript. Please find our responses below for general and minor comments.

*However, as the authors also pointed out, there are many unconsidered conditions like changes in TCs intensity, morphology, river discharge, etc. This undermines the value of particular resultant numbers if they are considered without more generalized conclusions. For example, what would happen if the SLR of 0.4 or 0.7m occurs? Some discussion going beyond the presented two SLR case studies would be appropriate. I propose to put the results described in the Section 3.4 in relation with the SLR values and not only with the present-day inundated area/storm surge height. It would be interesting to see the direct comparison (in % or absolute values) of changes in storm surge height with respect to SLR, e.g. for TC Sidr and Charchanda it would be 0.27m increase for 0.26m SLR, which is basically a linear addition of SLR magnitude on top of the present-day surge, and 0.62m change for 0.54m of SLR, which has a considerable non-linear contribution. This could give an insight into the (non)linearity of the storm surge and SLR interactions for particular area.*

Thank you for the suggestion. In section 3.4, we've made some revision in the calculated values of storm surge level and added two separate tables (Table 6 and Table 7) for both TC Sidr and TC Aila to show direct comparisons and percentage change of storm surge level under Sea Level Rise scenarios. Following are the corrected values in storm surge level:

Line 295 [Revised line 287]: 2.13 meters instead of 2.3 meters.

Line 296 [Revised line 288]: 13.7% instead of 21%.

Line 297 [Revised line 289]: 28.67% instead of 37%.

Line 298 [Revised line 290]: 2.41 m instead of 2.6 m.

Line 300 [Revised line 292]: 13.95% instead of 14% …. 1.87 meters instead of 2.24 meters……33.45% instead of 31%

Line 301 [Revised line 293]: 2.19 m instead of 2.59 m.

Line 302 [Revised line 296]: 21.93% instead of 22%..........1.299 meters instead of 1.61 meters.

Line 304 [Revised line 298]: 50.96% instead of 51%

Line 307 [Revised line 301]: 3.075 meters instead of 3.01 meters…….23% instead of 50%

Line 308 [Revised line 302]: 55% instead of 68%.

Based on the corrected calculations, we've also updated figure 7. In the initial submission, Figure 7a was mentioned as "TC Sidr at Barisal" and Figure 7b was mentioned as "TC Sidr at Charchanga". Actually, Figure 7a was representing TC Sidr at Charchanga and Figure 7b was representing TC Sidr at Barisal. We've corrected this mistake in the updated manuscript too. We've also added two additional table; one for the Tropical Cyclone tracks that were used for ensemble projections (Table 2) and one for the comparison of the change in storm surge inundated area (Table 5). Followings are the new Tables:

**Table 2** List of 12 historical TC events used for ensemble projection of storm surge inundation

| Name | Date | Landfall |
| --- | --- | --- |
| Tropical storm 13 | 14-18 November, 1973 | Noakhali |
| Cyclone 12 | 23-28 November, 1974 | Bhola |
| Tropical storm 19 | 07-12 November, 1975 | Chittagong |
| Tropical storm 1 | 22-25 May, 1985 | Noakhali |
| Cyclone 4 | 21-30 November, 1988 | Khulna |
| Cyclone 2 | 22-30 April, 1991 | Chittagong |
| Cyclone 2 | 26 April – 30 May, 1994 | Cox's Bazar |
| Cyclone 4 | 18-25 November, 1995 | Cox's Bazar |
| Cyclone 1 | 13-20 May, 1997 | Noakhali |
| Tropical storm 4 | 24-27 October, 2008 | Barguna |
| Tropical storm Mahasen | 10-16 May, 2013 | Patuakhali |
| Tropical storm Roanu | 18-21 May, 2016 | Chittagong |

**Table 5.** Comparison of inundated area between present day & future SLR scenarios and calculated change in percentage with respect to present day scenario.

| Scenario | TC Sidr | | TC Aila | |
|---|---|---|---|---|
| | Inundated Area | (%) change | Inundated Area | (%) change |
| Present Day | 1860 | | 1208 | |
| Mid-century | 2436.6 | +31 | 1550 | +28.3 |
| End-century | 2845.8 | +53 | 1770 | +46.5 |

**Table 6.** Comparison of storm surge level between present day and future SLR scenarios for the case of TC Sidr

| Scenario | Barisal | | Charchanga | |
|---|---|---|---|---|
| | Storm surge level (m) | % increase | Storm surge level (m) | % increase |
| Present Day | 1.873 | | 1.641 | |
| Mid-century (0.26m) | 2.13 | 13.72 | 1.870 | 13.95 |
| End-century (0.54m) | 2.41 | 28.67 | 2.19 | 33.45 |

**Table 7.** Comparison of storm surge level between present day and future SLR scenarios for the case of TC Aila

| Scenario | Barisal | | Charchanga | |
|---|---|---|---|---|
| | Storm surge level (m) | % increase | Storm surge level (m) | % increase |
| Present Day | 1.299 | | 2.5 | |
| Mid-century (0.26m) | 1.584 | 21.93 | 3.075 | 23 |
| End-century (0.54m) | 1.961 | 50.96 | 3.875 | 55 |

[Figure]

Figure 7. Comparison of storm surge water level between present day and future SLR scenarios. (a) TC Sidr at Barisal (b) TC Sidr at Charchanga (c) TC Aila at Barisal (d) TC Aila at Charchanga. The observed, modeled present-day, mid-of-21st century and end-of-21st century storm surge levels are denoted by the solid, red dashed, blue dotted, and greenred dash-dottted lines, respectively.

Regarding the relation between SLR and the additional increase of storm surge level following underlined paragraph was added at the end of section 3.4. Thank you for giving the idea to analyze this relation. [Revised line 304-314]:

"To analyze the linearity/non-linearity of storm surge level with respect to SLR, we conducted additional experiments based on 5 SLR scenarios; present-day sea level, 0.26 m of SLR, 0.33 m of SLR, 0.4 m of SLR, 0.47 m of SLR, 0.54 m of SLR, respectively. Results from these experiments are presented in the Figure 8.

For the case of TC Sidr in Barisal and Charchanga station, storm surge level increased almost linearly with respect to the addition of water due to the effect of SLR. For example, with an SLR of 0.47 m, the increase of storm surge level with respect to present day in Barisal and Charchanga stations were 0.453 m and 0.445 m, respectively (Figure 8a). On the other hand for the case of TC Aila, with an SLR of 0.26 m, the increase in storm surge level were found 0.285 m and 0.575 m respectively for the Barisal and Charchanga station (Figure 8b). Though the storm surge level is increasing almost linearly with the addition of sea water, however, there are still differences found between them, which could be influenced by the modification of ocean bathymetry to incorporate the effect of SLR. The margin of differences is higher for the Charchanga station comparing it with the Barisal station. The coarse resolution of topography near that area might be responsible for that."

[Figure]

**Figure 8.** Relation between SLR and increase in storm surge level with respect to the present-day simulated water level for TC Sidr and TC Aila. (a) is representing the relation for TC Sidr and (b) is representing the relation for TC Aila.

*It would be very helpful to include the terrain map of the model region with land elevation and land/water mask for better understanding of the present day situation and possible impacts. It could be combined with Figure 1 or not. Some names on Figure 1 would be also helpful (like main rivers, measurement location names, etc).*

We've added an additional figure showing the elevation of land and river as Figure 1b in the updated manuscript. The area near the coastal zone are very flat which can be seen from the figure 1b. This make the region vulnerable to flooding easily even under normal astronomical tide conditions. We've also shown the location of Ganges, Brahmaputra and Meghna river on Figure 1a. Two separate colors were used to represent the two observational stations. Thank you for the suggestion. Following is the updated Figure 1.

[Figure]

**Figure 1.** (a) Map of the study area for this work. The red and green lines represent the tracks of TC Sidr and TC Aila respectively. Area marked with green color indicates the Sundarban mangrove forest region. Two circles over the study area are the observation stations of Bangladesh Inland Water Transport Authority (BIWTA). The black colored outline shows the extent of model grid over the region. (b) Topography and bathymetry of the model domain. Negative depth values are representing water bodies (ocean and rivers) and positive depth values area representing land.

**Minor Comments:**

*- p2. lines 55-56: ". . .causing deaths . . . of lives". Please reformulate ('causing deaths of people' OR 'causing loss of lives')*

Corrected.

*- p3. line 83: please coordinate singular/plural forms "the impact . . . are debatable"*

Corrected.

*- p4. lines 110-114: this passage looks like repetition of the previous one. Please remove or reformulate.*

Removed.

*- p5. line 143: ". . . weas . . ." typo?*

Corrected.

*- p5. lines 151-171: It is not quite clear from this description whether Delft3D has special module for generating wind and pressure fields from the TC track. If yes, and it ". . . slightly improves the original WES. . .", why the authors are still using WES and not Delft3D? If not, and wind and pressure are firstly generated by WES method and then fed to the Delft3d, then the description is misleading.*

For storm surge simulations with Delft3D-FLOW, a Wind Enhance Scheme (WES) following Holland has been devised to generate tropical cyclone wind field (Delft Hydraulics, 2011). It's a built-in function in Delft3D-FLOW module. The earlier version of WES was developed by UK Met Office (Hemming et al. 1985). Later, it was further improved by Delft3D-FLOW by applying the translation speed of the cyclone center displacement as steering current and by introducing rotation of wind speed due to friction. All the version of WES is based on Holland's Wind Model (Holland, 1980). So, in this paper we used the improved version of WES (Delft Hydraulics, 2011) which is a built-in function of Delft3D-FLOW program. In the revised manuscript, we've added the citation of Delft3D-FLOW module's WES in section 2.1.3.

*- p6. line 193: ". . . Storm surge. . ." -> "storm surge"*

Corrected.

*- p7. line 209: by tidal data the water elevation is meant? What type of instrument has been used, tide gauge?*

By tidal data, we meant the water level elevation. Bangladesh Inland Water Transport Authority (BIWTA) uses auto tide gauge which provides hourly measurement of water level near the coast.

 *- p7. line 212: "(-ve)infinity . . ." typo?*

Actually by using (-ve)infinity, we meant negative. we've changed it to 'negative' in the revised version to remove the confusion.

*- p8. lines 269-272: repeated passage, please remove*

Removed. Thank you for pointing that out.

*- p10. lines 310-313: please review or remove the passage, it does not describe Figure 7.*

Removed.

*- p10. line 317: ". . . future TCs remain the same strength. . ." – either 'keep the same strength' or 'remain of the same strength'*

Corrected.

*- p10. line 322: "one of the methods we experimented in this study. . ." – either 'methods with which we experimented. . .' or 'methods that we tested'*

Corrected.

*- p10. line 327-328: ". . . the results looked much realistic. . ." -> . . . the results looked much more realistic. . .*

Corrected.

*- p10. Lines 344-345: ". . .focus of the paper is to predict the future scenarios. . ." – please change 'predict' to assess/estimate/develop ". . . and comparison with. . ." -> 'and to compare with'*

Corrected.

*- p11. line 360: ". . . first floor is kept transparent. . ." – What does transparent mean in this context? Why is it relevant here?*

To reduce confusion, this has been changed to "In TC shelters, the first floor should be kept above the high surge waters."

*- Figures 3b and 7a: for the case of TC Sidr in Charchanga the timeseries of measured and modelled water levels look somewhat different and out of phase. Do the authors have an explanation for this?*

I think here you mean 3b and 7b. For the case of TC Sidr in Charchanga station (Figure 3b and Figure 7b), there's a slight shift in phase occurred during the period of November 13 to November 15. This could be due to the presence of a seiche near the gauge that interfered with the astronomical tide, and the seiche was not resolved in the model simulation, which caused the discrepancy.

---

## Author Comment (AC3) · 24 Sep 2017

**Reply to Anonymous Referee #3**

*The paper uses numerical modelling to analyze the effects of sea level rise (SLR) on the storm surge generated by tropical cyclones (TC) in the Bangladesh coast and the associated inundation on that area. Model results are validated using observations of two previous TC and a number of additional simulations are made to study future scenarios. The manuscript is pretty well written, although it can be improved following the suggestions detailed below. Besides the assumptions made to simplify the high level of uncertainty, the obtained results show how SLR would increase the inundation associated to TC in this area and can help coastal managers to design adaptation measures to deal with these problems. Therefore, the manuscript fits the scope of NHESS and may be published provided the authors address the following comments.*

We would like to thank the referee for the evaluation and for providing feedback which helped to improve the manuscript. Please find our responses below for general and minor comments.

General comments –

*The authors should justify why they use the SLR projections from AR4 (IPCC, 2007) (line 60) instead of those from AR5 (IPCC, 2013), although they are based on the worst AR4 scenario (A1F1, line 72). Taking into account that regional SLR rates are much higher than the global rate (lines 64-66) and that global SLR projections from AR5 are worse than AR4, the scenarios considered by the authors could be too optimistic.*

The SLR projections used in this research is from Caesar et al. (2017; under review) which is based on IPCC AR5. We've modified the manuscript text to clarify and added the information about AR5.

In this proposed work, we will use the SLR projections from Caesar et al. (2017; under review), which **is based on IPCC AR5 and** suggests a projection of SLR of 26 cm for the mid-21st century (2040 -2060) and 54 cm for the end-21st century (2079 -2099).

*- A number of geographical sites are cited in the text (e.g. Bay of Bengal and Andaman Sea (line 37); Ganges, Brahmaputra, Meghna rivers (line 92); Baguna (lines 97 and 100); Patuakhali (lines 99 and 106); Khulna (lines 100 and 106); Jhalokati (line 100); Chandpur (line (106), Sundarban (lines 238, 239, 240)) that should be placed in a map to facilitate the reading of the text. In the same way, a figure showing the topography of the area would be very useful. In addition, the shorelines should be clearer in figures 1, 4, 5 and 8, to better understand the magnitude of the flooded areas.*

Thank you for the suggestion. We've updated Figure 1 and added another one with that to represent the topography and bathymetry of the study area as Figure 1b. We've made the shorelines more clear in the updated figures (Figure1, 4, 5, 8). The location of the Ganges, Brahmaputra and Meghna river was also marked on the map. Green colored area is showing Sundarban forest and its under Khulna division. In addition, the landfall locations of the storms were marked on the map. Since Patuakhali, Barguna and Jhalokathi are under the Barisal division, we've marked the Barisal division on the map. And Chandpur is under the Chittagong division and it was also marked on the map.

[Figure]

**Figure 1.** (a) Map of the study area for this work. The red and green lines represent the tracks of TC Sidr and TC Aila respectively. Area marked with green color indicates the Sundarban mangrove forest region. Location of the Ganges, Brahmaputra and Meghna rivers are shown on the map. Khulna, Barisal and Chittagong which are landfall locations for the historical TCs used for ensemble projection, shown inside a circular box on the map. Two circles over the study area are the observation stations of Bangladesh Inland Water Transport Authority (BIWTA). The black colored outline shows the extent of model grid over the region. (b) Topography and bathymetry of the model domain. Negative depth values represent water bodies (ocean and rivers) and positive depth values areas represent land.

*- In lines 201-206 the authors discuss the potential influence of the tide level on the inundation and indicate that different simulations have been performed considering diverse tide conditions, which are summarized in Table 2. However, nowhere is the magnitude of the tides shown. A description of tide features is necessary to understand the influence of this factor in the inundation.*

In line 200, we've added this:

For example, the tides shown in Figures 3 and 7 as the water level oscillations have amplitudes as high as 3 m, which could significantly affect the extension of flooded area, depending on whether the storm's landfall coincides with a high tide or a low tide.

And in line 203, we've added the following:

The change of timing in these tide-related experiments was implemented by modifiying the tracks of the storms so that their landfalls coincide with a high tide, a tide, or a zero-tide condition, in addition to their actual tidal phases.

*- The writing of sections 3.2, 3.3, 3.4 and 4 is a little bit confusing with the mixing of percentages, inundation areas and water levels. Perhaps the results could be summarized in a table to ease the understanding of the changes associated to each scenario.*

Thank you for the suggestion. We've added three new tables for section 3.3 and 3.4. in addition to that, we've added a new Table 2 for the TC tracks that were used in the ensemble projections.

**Table 5.** Comparison of inundated area between present day & future SLR scenarios and calculated change in percentage with respect to present day scenario.

| Scenario | TC Sidr | | TC Aila | |
|---|---|---|---|---|
| | Inundated Area | (%) change | Inundated Area | (%) change |
| Present Day | 1860 | | 1208 | |
| Mid-century | 2436.6 | +31 | 1550 | +28.3 |
| End-century | 2845.8 | +53 | 1770 | +46.5 |

**Table 6.** Comparison of storm surge level between present day and future SLR scenarios for the case of TC Sidr

| Scenario | Barisal | | Charchanga | |
|---|---|---|---|---|
| | Storm surge level (m) | % increase | Storm surge level (m) | % increase |
| Present Day | 1.873 | | 1.641 | |
| Mid-century (0.26m) | 2.13 | 13.72 | 1.870 | 13.95 |
| End-century (0.54m) | 2.41 | 28.67 | 2.19 | 33.45 |

**Table 7.** Comparison of storm surge level between present day and future SLR scenarios for the case of TC Aila

| Scenario | Barisal | | Charchanga | |
|---|---|---|---|---|
| | Storm surge level (m) | % increase | Storm surge level (m) | % increase |
| Present Day | 1.299 | | 2.5 | |
| Mid-century (0.26m) | 1.584 | 21.93 | 3.075 | 23 |
| End-century (0.54m) | 1.961 | 50.96 | 3.875 | 55 |

|  |  |  |  |  |
|--|--|--|--|--|
|  |  |  |  |  |

**Table 2** List of 12 historical TC events used for ensemble projection of storm surge inundation

| Name | Date | Landfall |
|------|------|----------|
| Tropical storm 13 | 14-18 November, 1973 | Noakhali |
| Cyclone 12 | 23-28 November, 1974 | Bhola |
| Tropical storm 19 | 07-12 November, 1975 | Chittagong |
| Tropical storm 1 | 22-25 May, 1985 | Noakhali |
| Cyclone 4 | 21-30 November, 1988 | Khulna |
| Cyclone 2 | 22-30 April, 1991 | Chittagong |
| Cyclone 2 | 26 April – 30 May, 1994 | Cox's Bazar |
| Cyclone 4 | 18-25 November, 1995 | Cox's Bazar |
| Cyclone 1 | 13-20 May, 1997 | Noakhali |
| Tropical storm 4 | 24-27 October, 2008 | Barguna |
| Tropical storm Mahasen | 10-16 May, 2013 | Patuakhali |
| Tropical storm Roanu | 18-21 May, 2016 | Chittagong |

*- Lines 321-324: In the discussion about the used methods, the authors say that they "included the increased sea level in open ocean boundary instead of adding it into the whole ocean depth". In my opinion this makes no sense because it introduces a discontinuity in the water level that physically is not possible. As the authors say, this produces an additional pressure gradient force acting towards the coast. Therefore, the obtained results are spurious. I suggest removing any reference to this method, including figure 8.*

This method was actually not used in this study to simulate future storm surge inundation. This method was used by some previous studies (Pickering et al. 2012) which we included in the revised text [Revised line 322]:

Rather, we included this as part of sensitivity experiments to show how the inundation could be different based on the consideration of SLR in the model. In the later part, we also clarified by mentioning this [Revised line 327-329]:

To make the future SLR simulation realistic, we considered the increased sea level in ocean bathymetry and increased the depth by 0.26 and 0.54 m, respectively, by considering land submergence near the coast. In that case, the result looked much more realistic than the previous one and this is the method we followed in this paper.

*Some of the presented results seem inconsistent:*

*o In Figure 5 the comparison of inundated areas between present day and future climate scenarios is shown. In this figure, there are several small areas of yellow color indicating zones flooded under present conditions but not flooded during future SLR conditions. The authors should explain why these low lying coasts are flooded with present SLR and not with higher SLR, contrary to what would be expected.*

We've added the following explanations regarding this in the manuscript [Revised line 258 – 261]

However in Figure 5, there are several small areas of yellow color indicating zones flooded under present conditions but not flooded during future SLR conditions. This is because Figure 5 showed snapshots of the inundation conditions at one particular time. Some areas may experience alternating wetting and drying conditions, which may explain why some areas are flooded with present SLR and not with higher SLR: this is so only at that particular time. The authors expect that those areas are flooded at other times.

*o In lines 226-227 the authors say: "the measured water level variation displayed larger amplitudes than did the model output". Observing Figure 3b, the trend seems the opposite (for positive values) and the red line (modeled) is located above the black one (observed). On the contrary, negative values and total oscillations are greater in the case of observed data. I suggest clarifying this point*

Thank you for pointing this out. We've updated the text [Revised line 222-224]

….the measured water level variation displayed smaller amplitudes than did the model output for positive tides and larger amplitudes than the modeled water level for negative tides, perhaps due to the coarse resolution of bathymetry.

*o When comparing water levels of Figure 7 and Figure 3, the observed and modelled values are different in panels (a) and (b) of both figures. It looks like in one of both figures, these panels are exchanged.*

We've made correction for calculated values of storm surge level in section 3.4 and relevant Figure 7.

Line 295 [Revised line 287]: 2.13 meters instead of 2.3 meters.

Line 296 [Revised line 288]: 13.7% instead of 21%.

Line 297 [Revised line 289]: 28.67% instead of 37%.

Line 298 [Revised line 290]: 2.41 m instead of 2.6 m.

Line 300 [Revised line 292]: 13.95% instead of 14% …. 1.87 meters instead of 2.24 meters……33.45% instead of 31%

Line 301 [Revised line 293]: 2.19 m instead of 2.59 m.

Line 302 [Revised line 296]: 21.93% instead of 22%..........1.299 meters instead of 1.61 meters.

Line 304 [Revised line 298]: 50.96% instead of 51%

Line 307 [Revised line 301]: 3.075 meters instead of 3.01 meters…….23% instead of 50%

Line 308 [Revised line 302]: 55% instead of 68%.

Based on the corrected calculations, we've also updated figure 7. In the initial submission, Figure 7a was mentioned as "TC Sidr at Barisal" and Figure 7b was mentioned as "TC Sidr at Charchanga". Actually, Figure 7a was representing TC Sidr at Charchanga and Figure 7b was representing TC Sidr at Barisal. We've corrected this mistake in the updated manuscript too.

[Figure]

Figure 7. Comparison of storm surge water level between present day and future SLR scenarios. (a) TC Sidr at Barisal (b) TC Sidr at Charchanga (c) TC Aila at Barisal (d) TC Aila at Charchanga. The observed, modeled present-day, mid-of-21st century and end-of-21st century storm surge levels are denoted by the black dash-dotted, red dotted, blue dashed, and green dash-dotted lines, respectively.

*Specific comments –*

*Lines 55-56: "the deaths of hundreds of thousands of lives". Better "the loss of hundreds of thousands of lives". This sentence is very similar to the following one: "This type of coastal flooding. . .", so probably both sentences could be combined into one.*

Thank you. We've updated the text and merged these lines [Revised line 54-56].

The geomorphological characteristics of the region have made the locale locales prone to major TC events, events which have occurred multiple times in the past, directly causing loss of hundreds of thousands of lives, property, livelihoods and the economy of the country (Haque, 1997).

*- Line 83: "The impact of climate change. . ... are still debatable" should be "The impact of climate change. . ... is still debatable" or "The impacts of climate change. . ... are still debatable".*

Thank you. It's corrected.

*- Line 88: "will be method of this study", better "will be the method of this study".*

Corrected.

*- The name of a district is written differently: Patuakhali (line 99), Patukhali (line 106), Pataukhali (line 106). Please be consistent and use only one name.*

Sorry about the mistakes. We've corrected these in the updated manuscript.

*- Lines 110-114: This paragraph seems a repetition of a previous one.*

Removed.

*- Lines 129-130: P0 and f are not defined in equations (2) and (3).*

It's now corrected. Thank you.

*- Line 156: The reference Heming et al. (1980) is missing or there is a mistake and should be Heming et al. (1995).*

It should be Heming et al. (1995). Thank you.

*- Line 167: The meaning of e is not defined in equation (6).*

It's the base of the natural logarithm (=2.71828182846) (Delft Hydraulics, 2011). We've included this information in the revised version.

*- Line 178: "methods described in Zhang et al. (2012) was followed" should be "methods described in Zhang et al. (2012) were followed".*

Corrected.

*- Line 184: "boundary was shown in Figure 1", better "boundary is shown in Figure 1". - Line 206: ". . .in making ensemble projections shown in Table 2" should be ". . .in making ensemble projections are shown in Table 2"*

Corrected.

*- Line 212: "(-ve)" looks a typo.*

It's now corrected.

*- Line 215: Equation (8) is wrong. The MAE is obtained by comparing observations with model results*

Corrected.

*- Line 234: "the two TCs considered were shown in Figure 4.", better : "the two TCs considered are shown in Figure 4."*

Thank you. We've corrected it.

*- Lines 262-263: Please substitute "square kilometers" by "km2".*

Corrected.

*- Lines 269-272: This paragraph is a repetition of the previous one.*

Thanks for pointing this out. We've removed those lines.

*- Lines 310-313 are redundant with the previous paragraphs and although they coincide with Figure 7 caption (which is wrong), they do not describe Figure 7.*

Sorry about the mistake. We've removed that paragraph and corrected the caption of Figure 7

*- Line 351: "SLR conditions; which is. . .", better "SLR conditions, which is. . .".*

Thank you. We've corrected this.

*- References: Alam (1996), Mohal et al. (2006) and Vatvani et al. (2002) are listed in References but are not cited in the text.*

Alam (1996) and Mohal et al. (2006) were removed from the updated manuscripts. Vatvani et al. (2002) should be in line 168. In addition to that, references regarding WES (Delft Hydraulics, 2011) was added in line 156 and was listed in references list.

*- The reference corresponding to Delft3D model is cited in the text as Delft Hydraulics (2006) but is listed as Hydraulics, D. (2006). Please be consistent.*

Sorry about that. We've updated that.

*- Figure 7 caption is wrong and it does not describe this figure, since the results of both future scenarios are included in each figure.*

Caption is now updated as follows [Revised line 538 - 540]

Figure 7. Comparison of storm surge water level between present day and future SLR scenarios. (a) TC Sidr at Barisal (b) TC Sidr at Charchanga (c) TC Aila at Barisal (d) TC Aila at Charchanga. The observed, modeled present-day, mid-of-21st century and end-of-21st century storm surge levels are denoted by the black dash-dotted, red dotted, blue dashed, and green dash-dotted lines, respectively.

---

## Author Comment (AC4) · 24 Sep 2017

[revised manuscript text omitted]

**Figure 9.** Comparison of inundated areas for TC Aila between present-day and end-21$^{st}$-century (0.54m SLR) scenario. White color is
representing the increased flooded areas that were not in present-day scenario but the increase due to future SLR. Red color is showing the
inundated area that's similar both for present-day and future scenario case. Blue areas are either land or constant waters (those which are already
water at the model initialization time). Figure (a) is representing the inundated area when SLR was considered on ocean depths instead of adding
it in to the open ocean boundary and Figure (b) is showing the inundated area when we considered the SLR on ocean boundary.

**Table 1** Manning's Roughness Coefficient for different land coverings.

| Land cover | Manning's coefficient |
|---|---|
| River | 0.015 |
| Mangrove | 0.080 |
| Ocean | 0.01 |
| Land | 0.025 |

**Table 2** List of 12 historical TC events used for ensemble projection of storm surge inundation

| Name | Date | Landfall location |
|---|---|---|
| Tropical storm 13 | 14-18 November, 1973 | Noakhali |
| Cyclone 12 | 23-28 November, 1974 | Bhola |
| Tropical storm 19 | 07-12 November, 1975 | Chittagong |
| Tropical storm 1 | 22-25 May, 1985 | Noakhali |
| Cyclone 4 | 21-30 November, 1988 | Khulna |
| Cyclone 2 | 22-30 April, 1991 | Chittagong |
| Cyclone 2 | 26 April – 30 May, 1994 | Cox's Bazar |
| Cyclone 4 | 18-25 November, 1995 | Cox's Bazar |
| Cyclone 1 | 13-20 May, 1997 | Noakhali |
| Tropical storm 4 | 24-27 October, 2008 | Barguna |
| Tropical storm Mahasen | 10-16 May, 2013 | Patuakhali |
| Tropical storm Roanu | 18-21 May, 2016 | Chittagong |

**Table 3:** Parameters considered for ensemble projection of storm surge inundation which includes the TC intensities, tidal conditions and the SLR scenarios.

| TC name | Intensities | Tide conditions | SLR |
|---|---|---|---|
| TC Sidr | +10%, present day, -10% | High Tide, low tide, actual tide, zero tide | Present day, 0.26 m, 0.54 m |
| TC Aila | +10%, Present day, -10% | High Tide, low tide, actual tide, zero tide | Present day, 0.26 m, 0.54 m |
| 12 historical TC tracks | Actual intensities | Actual tide conditions | Present day, 0.26 m, 0.54 m |

**Table 4.** Computed values of RMSE, MAE and Nash-Sutcliffe coefficient for both TC Sidr and TC Aila

| Stations | TC Sidr | | | TC Aila | | |
|---|---|---|---|---|---|---|
| | RMSE (m) | MAE (m) | NASH | RMSE (m) | MAE (m) | NASH |
| Barisal | 0.23 | 0.16 | 0.85 | 0.33 | 0.24 | 0.65 |
| Charchanga | 0.26 | 0.19 | 0.80 | 0.28 | 0.17 | 0.73 |

**Table 5.** Comparison of inundated area between present-day & future SLR scenarios and calculated change in percentage with respect to present-day scenario.

| Scenario | TC Sidr | | TC Aila | |
|---|---|---|---|---|
| | Inundated Area (km$^2$) | (%) increase | Inundated Area (km$^2$) | (%) increase |
| Present-day | 1860 | | 1208 | |
| Mid-21$^{st}$-century | 2436.6 | +31 | 1550 | +28.3 |
| End-21$^{st}$-century | 2845.8 | +53 | 1770 | +46.5 |

**Table 6.** Comparison of storm surge level between present-day and future SLR scenarios for the case of TC Sidr

| Scenario | Barisal | | Charchanga | |
|---|---|---|---|---|
| | Storm surge level (m) | % increase | Storm surge level (m) | % increase |
| Present-day | 1.873 | | 1.641 | |
| Mid-21$^{st}$-century (0.26m) | 2.13 | 13.72 | 1.870 | 13.95 |
| End-21$^{st}$-century (0.54m) | 2.41 | 28.67 | 2.19 | 33.45 |

**Table 7.** Comparison of storm surge level between present-day and future SLR scenarios for the case of TC Aila

| Scenario | Barisal | | Charchanga | |
|---|---|---|---|---|
| | Storm surge level (m) | % increase | Storm surge level (m) | % increase |
| Present-day | 1.299 | | 2.5 | |
| Mid-21$^{st}$-century (0.26m) | 1.584 | 21.93 | 3.075 | 23 |
| End-21$^{st}$-century (0.54m) | 1.961 | 50.96 | 3.875 | 55 |

---

## Referee Report (RR1)

Dear authors, thank you for reworking the manuscript. It became more understandable with additional figures and tables.

There are several minor points, which on my opinion still should be clarified:

1. Am I right that Figure 3 and Figure 7 "Present day" are the results from the same "real TC" modeling? In this case, please check Fig.7d, because it does not correspond to Fig.3d. Namely, at Fig.3d modeled water levels overestimate observations consistently, at Fig.7d modeled water levels underestimate observations and fit for the storm peak. Which one is correct? Moreover, for TC Aila the dates at Fig.3 (c,d) and Fig.7(c,d) are shifted by one day.

2. Table 6: in the sensitivity experiment, it still would be interesting to see the change as percentage of applied SLR. What the authors showed is the changes as percentage of "present day" water level. This is also interesting to see, however, this does not say much about (non)linearity of surge and SLR interactions. If one looks at changes as % of SLR, an interesting pattern shows up: for TC Sidr and both considered locations the water levels are almost linear addition of "present day" levels and SLR (changes between 88% and 103%). For TC Aila the surge/SLR interaction becomes non-linear, the changes there are about 111%-124% for Barisal and 219%-253% (!) for Charchanga. Why the same SLR for the same location (say Chrachanga) causes such different changes for two different TCs? I'm not expecting the authors to answer this question in the paper, but it is interesting.

3. In Section 3.4, caption of Figure 7, caption and text of Table 6 the authors write "storm surges". However, in reality it is total water level that is discussed and not storm surge (which is normally defined as difference between total water levels and normal tides). Please reformulate throughout the text or describe what is understood under "storm surge" in this paper, in the present form it is misleading.

4. p9. Line 296: typo – "present-day" two times

---

## Author Response (AR2)

Dear Editor Dr. Piero Lionello and Referees,

Thank you for all the comments and suggestions you made so far which helped to significantly improve the manuscript from its initial phase. We've corrected the errors and revised it one more time based on the comments. Please find our responses to the comments and suggestions below.

**Response to Editor's Comments:**

*Dear Authors,*

*The reviewers have commented on the new version of your manuscript. They agree that you have made substantial efforts to address their comments and have properly addressed the points arisen during the first review, but few points still need to be clarified.*

*I am reporting below their comments. Your manuscript will not be returned to the reviewers, but I will check how you have modified it according to their suggestions or replied to their comments. Please return with the revised manuscript also a point-to-point list of changes corresponding to the list below.*

*Do not hesitate to contact me if anything is not clear*

*Piero Lionello*

Dear Dr. Piero Lionello,

Thank you for the suggestions and combining the comments from all the referees.

Please find below the responses to the points mentioned and the marked up version of the manuscript.

*1. In Figure 3 and Figure 7 "Present day" should be the results of the same "real TC" modeling. If this is correct, please check Fig.7d, because it does not correspond to Fig.3d. Namely, at Fig.3d modeled water levels overestimate observations consistently, at Fig.7d modeled water levels underestimate observations and fit for the storm peak. Which one is correct? Moreover, for TC Aila the dates at Fig.3 (c,d) and Fig.7(c,d) are shifted by one day.*

We apologize for the mistake. Figure 3 which was used for validation is actually the correct one. We've corrected the scale in Figure 7 c, d. Also, in Figure 7.d, we've corrected the water levels for the Present Day (Observed) and Present Day (Modeled). Thank you for pointing this out.

[Figure]

**Figure 7.** Comparison of storm surge water levels between present-day and future SLR scenarios. (a) TC Sidr at Barisal (b) TC Sidr at Charchanga (c) TC Aila at Barisal (d) TC Aila at Charchanga. The observed, modeled present-day, mid-21st-century and end-21st-century storm surge levels are denoted by the black dash-dotted, red dotted, blue dashed, and green dash-dotted lines, respectively.

*2. Table 6: in the sensitivity experiment, please include the change as percentage of applied SLR. What the table now includes is the changes as percentage of "present day" water level. This is also interesting to see, however, this does not say much about (non)linearity of surge and SLR interactions. If one looks at changes as % of SLR, an interesting pattern shows up: for TC Sidr and both considered locations the water levels are almost linear addition of "present day" levels and SLR (changes between 88% and 103%). For TC Aila the surge/SLR interaction becomes non-linear, the changes there are about 111%-124% for Barisal and 219%-253% (!) for Charchanga. Why the same SLR for the same location (say Chrachanga) causes such different changes for two different TCs? Could you comment on this?*

Thank you for the suggestion. We've updated Table 6 and included the percentage change in peak water level with respect to the SLR change. We've also updated the text in line 298 – 304.

The table shows almost linear increase for TC Sidr with a range between 88% to 103%. For TC Aila in Barisal station, the increase was with in 112% to 124% which is slightly higher than SLR. But for TC Aila in Charchanga station, the increase is significantly higher than SLR.

This could be due to the coarse resolution of bathymetry and the location of the stations. Some of the points that was mentioned in the validation chapter in section 3.1 can also be considered for this case. In line 212-213 we mentioned about the coarse resolution over Charchanga and location of these two-observational station. **"Barisal station is located more towards the inland whereas Charchanga is located near the coastline where the grid cell resolution was coarse. But none of them are in the open ocean water, which is important to get a clear idea about peak water level."** So, the coarse resolution can also be a factor as well as their location. In theory, in the open coastal water, the increase in peak water level due to SLR should approximately match the SLR, as suggested by the simplified equation $\Delta\eta = \frac{\tau_w L}{g\rho h}$ in the *Introduction* section. However, the two stations used in the paper are both located away from the open water, and thus the complex topography and bathymetry may have played a role in regulating the peak water levels at these two particular locations in ways that are different than if they had been in the open coastal water.

**Table 6.** Comparison of storm surge level between present day and future SLR scenarios and increase in storm surge level with respect to the present day and SLR scenario for the case of TC Sidr and TC Aila in Barisal and Charchanga observational stations. The SLR scenarios of 0.33 m, 0.40 m and 0.47 m were used to examine the linearity/non-linearity of increase in storm surge level with respect to SLR conditions. In the table "w.r.t" stands for "with respect to".

| SLR (m) | TC Sidr | | | | | | TC Aila | | | | | |
| | Barisal | | | Charchanga | | | Barisal | | | Charchanga | | |
| | surge(m) | increase w.r.t present-day (m) and (%) | increase w.r.t. SLR (%) | surge(m) | increase w.r.t present day (m) and (%) | increase w.r.t. SLR (%) | surge (m) | increase w.r.t. present day (m) and (%) | increase w.r.t. SLR (%) | surge (m) | increase w.r.t present day (m) and (%) | increase w.r.t. SLR (%) |
|---|---|---|---|---|---|---|---|---|---|---|---|---|
| 0.00 | 1.87 | n/a | n/a | 1.64 | n/a | n/a | 1.29 | n/a | n/a | 2.50 | n/a | n/a |
| 0.26 | 2.13 | 0.26 (14%) | 100% | 1.87 | 0.23 (14%) | 88% | 1.58 | 0.29 (22%) | 112% | 3.07 | 0.57 (23%) | 219% |
| 0.33 | 2.21 | 0.34 (18%) | 103% | 1.95 | 0.31 (19%) | 94% | 1.66 | 0.37 (29%) | 112% | 3.22 | 0.72 (29%) | 218% |
| 0.40 | 2.26 | 0.39 (21%) | 97% | 2.00 | 0.36 (22%) | 90% | 1.75 | 0.46 (36%) | 115% | 3.42 | 0.92 (37%) | 230% |
| 0.47 | 2.32 | 0.45 (24%) | 96% | 2.08 | 0.44 (27%) | 94% | 1.82 | 0.53 (41%) | 113% | 3.67 | 1.17 (47%) | 249% |
| 0.54 | 2.41 | 0.54 (29%) | 100% | 2.19 | 0.55 (34%) | 102% | 1.96 | 0.67 (52%) | 124% | 3.87 | 1.37 (55%) | 254% |

*3. In Section 3.4, caption of Figure 7, caption and text of Table 6 you write "storm surges". However, in reality it is total water level that is discussed and not storm surge (which is normally defined as difference between total water levels and normal tides). Please reformulate throughout the text or describe what is understood under "storm surge" in this paper, in the present form it is misleading.*

Thank you for the suggestion. We agree with what you mentioned. We have added the following footnote on page 4 to clarify.

[1] In this paper, the storm surge results refer to the total water level including both normal tides and the water level change due to storm

*4. - Lines 254-255: it is not clear from the caption of Figure 5 that the maps showing inundated areas are relative to a specific time instead of depicting the total extent of the TC-induced flooding. Actually, to show the total extent of the TC-induced flooding would make more sense also for consistency with the values reported in Table 5. Could this be changed?*

Thank you for the suggestion. The inundation map is showing the snapshots with the overall near-maximum extent of flooded area which was found during the landfall time for each model run. The model output data have an interval of 6 hours. So, at the next model output time, 6 hours after the landfall, the water started

receding and as a result, inundated area also started decreasing. Therefore, the areas shown in Figure 5 are the near maximum range of inundated area for the events, although with some exceptions shown in Figure 5 when some small areas are flooded in present-day condition but not in SLR scenarios. However, these exceptions do not affect the overall conclusion.

Minor typos and rewording:

- Line 47: "Harris et al., 1963". According to the reference list it should be cited as "Harris, 1963".

Corrected.

- Line 106: "storm surge inundation were discussed". Better "storm surge inundation are discussed".

Corrected.

- Line 109: "storm surge level were presented". Better "storm surge level are presented".

Corrected.

- Line 126: please capitalize "coriolis".

Corrected.

- Line 129: "Fξ are the turbulent momentum flux" should be "Fξ is the turbulent momentum flux".

Corrected.

 Line 129: "Fη are the turbulent momentum flux" should be "Fη is the turbulent momentum flux".

Corrected.

- Line 175: "Esgbert et al. (1994) Location of the…" should be "Esgbert et al. (1994). Location of the…".

Corrected.

- Lines 218-219: the explanation of Figure 3b is confusing. Maybe the words "and for negative tides" are to be expunged.

Removed.

- Line 296: typo – "present-day" two times

Corrected

- Line 299: "with an SLR of 0.47 m" should be "with a SLR of 0.47 m".

Corrected. We've also replaced 'an' with 'a' before SLR throughout the whole text.

- Line 366: "under the condition of 0.26 m and 0.54 m respectively", better "under SLR of 0.26 m and 0.54 m respectively".

Corrected

- Line 452: please replace "shown" with "are shown".

Corrected.

- In Table 2, information about the intensity of the considered TCs would also be interesting, such as maximum wind speed in the considered area or SLP minimum at landfall

Thank you for the suggestion. We've added a separate column showing the maximum sustained wind speed for each of those storms used in ensemble projections.

- In Table 5, the number of significant digits is excessive considering the uncertainties of the results. Integer values would be more appropriate. For the same reason, in Table 6 percent values should be rounded to integers. Please also change the text accordingly.

Thank you. We've updated the table and updated the texts accordingly.

**Response to Anonymous Referee #1**

*The authors made substantial efforts to address the Reviewers' comments and improve clarity of the whole manuscript, with particular regard to the presentation of results. However, a few issues still need some clarification. Therefore, in my opinion, the paper can be accepted for publication with minor revision.*

We appreciate the comments from the referee and would like to thank for evaluations and feedback which helped to improve the manuscript.

*Specific comments:*

- *Line 126: please capitalize "coriolis".*

Thank you for noticing this. We've Corrected it.

- *Lines 218-219: the explanation of Figure 3b is confusing. Maybe the words "and for negative tides" are to be expunged, if I am not misunderstood.*

Thank you for the suggestion. We've removed that.

- *Lines 254-255: it is not clear from the caption of Figure 5 that the maps showing inundated areas are relative to a specific time instead of depicting the total extent of the TC-induced flooding, which would make more sense also for consistency with the values reported in Table 5.*

Thank you for the suggestion. The inundation map is showing the snapshots with the overall near-maximum extent of flooded area which was found during the landfall time for each model run. The model output data have an interval of 6 hours. So, at the next model output time, 6 hours after the landfall, the water started receding and as a result, inundated area also started decreasing. Therefore, the areas shown in Figure 5 are the near maximum range of inundated area for the events, although with some exceptions shown in Figure 5 when some small areas are flooded in present-day condition but not in SLR scenarios. However, these exceptions do not affect the overall conclusion.

- *Line 452: please replace "shown" with "are shown".*

Corrected.

- *In Table 2, information about the intensity of the considered TCs would also be interesting.*

Thank you for the suggestion. We've added a separate column showing the maximum sustained wind speed for each of those storms used in ensemble projections.

- *In Table 5, the number of significant digits is excessive considering the uncertainties of the results. Integer values would be more appropriate. For the same reason, in Table 6 percent values should be rounded to integers. Please also change the text accordingly.*

Thank you. We've updated the table and updated the texts accordingly.

Thank you again for all the suggestions and corrections.

**Response to Anonymous Referee #2**

*Dear authors, thank you for reworking the manuscript. It became more understandable with additional figures and tables.*

Thank you Referee for your valuable comments and suggestions which helped to improve the manuscript from its initial phase.

*There are several minor points, which on my opinion still should be clarified:*

*1. Am I right that Figure 3 and Figure 7 "Present day" are the results from the same "real TC" modeling? In this case, please check Fig.7d, because it does not correspond to Fig.3d. Namely, at Fig.3d modeled water levels overestimate observations consistently, at Fig.7d modeled water levels underestimate observations and fit for the storm peak. Which one is correct? Moreover, for TC Aila the dates at Fig.3 (c,d) and Fig.7(c,d) are shifted by one day.*

We apologize for the mistake. Figure 3 which was used for validation is actually the correct one. We've corrected the scale in Figure 7 c, d. Also, in Figure 7.d, we've corrected the water levels for the Present Day (Observed) and Present Day (Modeled). Thank you for pointing this out.

[Figure]

**Figure 7.** Comparison of storm surge water levels between present-day and future SLR scenarios. (a) TC Sidr at Barisal (b) TC Sidr at Charchanga (c) TC Aila at Barisal (d) TC Aila at Charchanga. The observed, modeled present-day, mid-21st-century and end-21st-century storm surge levels are denoted by the black dash-dotted, red dotted, blue dashed, and green dash-dotted lines, respectively.

*2. Table 6: in the sensitivity experiment, it still would be interesting to see the change as percentage of applied SLR. What the authors showed is the changes as percentage of "present day" water level. This is also interesting to see, however, this does not say much about (non)linearity of surge and SLR interactions. If one looks at changes as % of SLR, an interesting pattern shows up: for TC Sidr and both considered locations the water levels are almost linear addition of "present day" levels and SLR (changes between 88% and 103%). For TC Aila the surge/SLR interaction becomes non-linear, the changes there are about 111%-124% for Barisal and 219%-253% (!) for Charchanga. Why the same SLR for the same location (say Chrachanga) causes such different changes for two different TCs? I'm not expecting the authors to answer this question in the paper, but it is interesting.*

Thank you for the suggestion. We've updated Table 6 and included the percentage change in peak water level with respect to the SLR change. We've also updated the text in line 298 – 304.

The table shows almost linear increase for TC Sidr with a range between 88% to 103%. For TC Aila in Barisal station, the increase was with in 112% to 124% which is slightly higher than SLR. But for TC Aila in Charchanga station, the increase is significantly higher than SLR.

This could be due to the coarse resolution of bathymetry and the location of the stations. Some of the points that was mentioned in the validation chapter in section 3.1 can also be considered for this case. In line 212-213 we mentioned about the coarse resolution over Charchanga and location of these two-observational station. **"Barisal station is located more towards the inland whereas Charchanga is located near the coastline where the grid cell resolution was coarse. But none of them are in the open ocean water, which is important to get a clear idea about peak water level."** So, the coarse resolution can also be a factor as well as their location. In theory, in the open coastal water, the increase in peak water level due to SLR should approximately match the SLR, as suggested by the simplified equation $\Delta\eta = \frac{\tau_w L}{g\rho h}$ in the *Introduction* section. However, the two stations used in the paper are both located away from the open water, and thus the complex topography and bathymetry may have played a role in regulating the peak water levels at these two particular locations in ways that are different than if they had been in the open coastal water.

**Table 6.** Comparison of storm surge level between present day and future SLR scenarios and increase in storm surge level with respect to the present day and SLR scenario for the case of TC Sidr and TC Aila in Barisal and Charchanga observational stations. The SLR scenarios of 0.33 m, 0.40 m and 0.47 m were used to examine the linearity/non-linearity of increase in storm surge level with respect to SLR conditions. In the table "w.r.t" stands for "with respect to".

| SLR (m) | TC Sidr | | | | | | TC Aila | | | | | |
| --- | --- | --- | --- | --- | --- | --- | --- | --- | --- | --- | --- | --- |
| | Barisal | | | Charchanga | | | Barisal | | | Charchanga | | |
| | surge(m) | increase w.r.t present-day (m) and (%) | increase w.r.t. SLR (%) | surge(m) | increase w.r.t present day (m) and (%) | increase w.r.t. SLR (%) | surge (m) | increase w.r.t. present day (m) and (%) | increase w.r.t. SLR (%) | surge (m) | increase w.r.t present day (m) and (%) | increase w.r.t. SLR (%) |
| 0.00 | 1.87 | n/a | n/a | 1.64 | n/a | n/a | 1.29 | n/a | n/a | 2.50 | n/a | n/a |
| 0.26 | 2.13 | 0.26 (14%) | 100% | 1.87 | 0.23 (14%) | 88% | 1.58 | 0.29 (22%) | 112% | 3.07 | 0.57 (23%) | 219% |
| 0.33 | 2.21 | 0.34 (18%) | 103% | 1.95 | 0.31 (19%) | 94% | 1.66 | 0.37 (29%) | 112% | 3.22 | 0.72 (29%) | 218% |
| 0.40 | 2.26 | 0.39 (21%) | 97% | 2.00 | 0.36 (22%) | 90% | 1.75 | 0.46 (36%) | 115% | 3.42 | 0.92 (37%) | 230% |
| 0.47 | 2.32 | 0.45 (24%) | 96% | 2.08 | 0.44 (27%) | 94% | 1.82 | 0.53 (41%) | 113% | 3.67 | 1.17 (47%) | 249% |
| 0.54 | 2.41 | 0.54 (29%) | 100% | 2.19 | 0.55 (34%) | 102% | 1.96 | 0.67 (52%) | 124% | 3.87 | 1.37 (55%) | 254% |

*3. In Section 3.4, caption of Figure 7, caption and text of Table 6 the authors write "storm surges". However, in reality it is total water level that is discussed and not storm surge (which is normally defined as difference between total water levels and normal tides). Please reformulate throughout the text or describe what is understood under "storm surge" in this paper, in the present form it is misleading.*

Thank you for the suggestion. We agree with what you mentioned. We have added the following footnote on page 4 to clarify.

[1] In this paper, the storm surge results refer to the total water level including both normal tides and the water level change due to storm

*4. p9. Line 296: typo – "present-day" two times*

Corrected.

Thank you again for all the suggestions and corrections.

[revised manuscript text omitted]

---

## Author Response (AR3)

Response to Editor's Comment:

Dear Authors:

thanks for your work at the revised manuscript and for having answered the comments of the reviewers. However, before accepting your manuscript for publication, I am forced to ask you a further revision.

It appears now from the added information in table 6 that for TC Aila at Charchanga, the sea level increase during the storm greatly exceeds (by a factor larger than 200%) the simple effect of sea level rise.

At this point, your lines 303-306 are not clear.

First, you cannot maintain that "Though the storm surge level is increasing almost linearly with the addition of sea water", as percent deviations from SLR are huge in one location out of four. Further the following 3 lines do not really explain why there is this large deviation in Charchanga and whether you think this 200% increase is realistic or not. Finally these lines are not clear: what is meant with "them" in line 304 and with "that" in line 306? Is left to the readers to be guessed. Please reformulate lines 303-306.

Looking forward to receiving your final manuscript.

Piero Lionello

Dear Editor Dr. Piero Lionello,

Thank you're the suggestions that helped us to make the manuscript better.

Regarding lines 303 – 306: We agree with what you mentioned and made the following corrections to improve the explanations.

(a) We've removed the lines 303-306.

(b) Following lines were now added in place of line 303-306

*The increase in storm surge level for TC Aila in Charchanga station is significantly higher than what we found in the Barisal station (Table 6). This discrepancy could be due to the coarse resolution of topography and bathymetry near the stations. In theory, in open coastal water, the increase in peak water level due to SLR should approximately match the SLR, as suggested by the simplified equation $\Delta \eta = \frac{\tau_w L}{g \rho h}$ in in the Introduction section. However, the two stations used in the paper are both located away from the open water, and thus the complex topography and bathymetry may have played a role in regulating the peak water levels at the locations in ways that are different than if they had been in the open coastal water. In other words, the simulated 219% storm surge level increase for TC Aila at the Charchanga station may not realistic; instead, it could be attributed to artificial effects due to coarse resolution of the topography and bathymetry near the station.*

Thank you again to you and all the reviewers for their comments and suggestions.

Regards,

Mansur Ali Jisan

[revised manuscript text omitted]

---

## Author Response (AR4)

Comments to the Author:

Dear Mansur Ali Jisan ,

let me suggest to slightly rephrase your text:

"The increase in storm surge level for TC Aila in Charchanga station is significantly higher than what we found in the Barisal station (Table 6). This difference is likely due to the different role played by the topography and bathymetry near the two stations and their coarse resolution in model grid. In absence of a strong interaction with the bottom topography, the increase in peak water level during the hurriccanes should approximately match the mean SLR. However, both Charchanga and Barisal are located away from the open water, and thus the complex topography and bathymetry play a role in regulating water levels in a complex way. The large simulated storm surge level increase for TC Aila at the Charchanga station may be unrealistic and in our opinion it is an artificial effect due to coarse resolution of the model grid near the station. "

Obviouly feel free changing it.

Otherwise, if you agree, please submit a new version with these new phrasing.

Best regards,

Piero

Dear Dr. Piero Lionello,

Thank you for the suggestions. We agree with what you mentioned. We've updated the text accordingly.

Best Regards,

Mansur

[revised manuscript text omitted]